



Ubiquitous production of branched glycerol dialkyl glycerol tetraethers (brGDGTs) in
global marine environments: a new source indicator for brGDGTs
Wenjie Xiao[1,2], Yinghui Wang[2], Shangzhe Zhou[2], Limin Hu[3], Huan Yang[4], Yunping Xu[1,2*]
[1]Shanghai Engineering Research Center of Hadal Science and Technology, College of Marine
Sciences, Shanghai Ocean University, Shanghai 201306, China
[2]MOE Key Laboratory for Earth Surface Process, College of Urban and Environmental Sciences,
Peking University, Beijing 100871, China
[3]Key Laboratory of Marine Sedimentology and Environmental Geology, First Institute of
Oceanography, State Oceanic Administration, Qingdao 266061, China
[4]State Key Laboratory of Biogeology and Environmental Geology, China University of Geosciences,
Wuhan 430074, China
Corresponding author: Y Xu (ypxu@shou.edu.cn)
Abstract. Presumed source specificity of branched and isoprenoid glycerol dialkyl
glycerol tetraethers (GDGTs) led to the development of several biomarker proxies for
biogeochemical cycle and paleoenvironment. However, recent studies reveal that
brGDGTs are also produced in aquatic environments besides soils and peat. Here we
examined three cores from the Bohai Sea and found distinct difference in brGDGT
compositions varying with the distance from the Yellow River mouth. We thus proposed
an abundance ratio of hexamethylated to pentamethylated brGDGT (IIIa/IIa) to
evaluate brGDGT sources. The compiling of globally distributed 1354 marine
sediments and 589 soils shows that the IIIa/IIa ratio is generally <0.59 for soils, 0.59–
0.92 and >0.92 for marine sediments with and without significant terrestrial inputs,
respectively. Such disparity confirms the existence of two sources of brGDGTs, a
terrestrial origin with lower IIIa/IIa and a marine origin with higher IIIa/IIa, likely due
to different pH influence. The application of the IIIa/IIa ratio to the East Siberian Arctic
Shelf proves it a sensitive source indicator for brGDGTs, which is helpful for accurate





estimation of organic carbon source and paleoclimates in marine settings.
1 Introduction

31         Glycerol dialkyl glycerol tetraethers (GDGTs), membrane lipids of archaea and

certain bacteria, are widely distributed in marine and terrestrial environments
(Reviewed by Schouten et al., 2013). These lipids have become a focus of attention of
organic geochemists for more than ten years because they can provide useful
environmental and climatic information such as temperature, soil pH, organic carbon
source and microbial community structure (e.g., Hopmans et al., 2004; Kim et al., 2010;
Lipp et al., 2008; Peterse et al., 2012; Schouten et al., 2002; Weijers et al., 2006; Zhu
et al., 2016). There are generally two types of GDGTs, isoprenoid (iGDGTs) and non-
isoprenoid, branched GDGTs (brGDGTs; Fig. 1). The former group is more abundant
in aquatic settings and generally thought to be produced by Thaumarchaeota, a specific
genetic cluster of the archaea domain (Schouten et al., 2008; Sinninghe Damsté et al.,
2002), although Euryarchaeota may be a significant source of iGDGTs in the ocean
(e.g., Lincoln et al., 2014). In contrast, brGDGTs having 1,2-di-*O*-alkyl-*sn*-glycerol
configuration are substantially more abundant in peat and soils than marine sediments,
supporting that they are derived from bacteria rather than archaea (Sinninghe Damsté
et al., 2000; Weijers et al., 2006). So far, only two species of Acidobacteria were
identified to contain one brGDGT with two 13,16-dimethyl octacosanyl moieties
(Sinninghe Damsté et al., 2011), which is contrast to high diversity and ubiquitous
occurrence of a series of brGDGTs with four to six methyl groups and zero to two
cyclopentane rings in environments (Weijers et al., 2007b). Therefore, other biological
sources of brGDGTs, although not yet identified, are likely.

52         The source difference between brGDGTs and iGDGTs led researchers to

developing a branched and isoprenoid tetraether (BIT) index, expressed as relative
abundance of terrestrial-derived brGDGTs to aquatic-derived crenarchaea (Hopmans et
al., 2004). Subsequent studies found that the BIT index is specific for soil organic
carbon because GDGTs are absent in vegetation (e.g., Sparkes et al., 2015; Walsh et al.,





2008). The BIT index is generally higher than 0.9 in soils, but close to 0 in marine
sediments devoid of terrestrial inputs (Weijers et al., 2006). Since its advent, the BIT
index has been increasingly used in different environments (e.g., Blaga et al., 2011;
Herfort et al., 2006; Kim et al., 2006; Loomis et al., 2011; Wu et al., 2013). Besides the
BIT index, Weijers et al. (2007b) found that the number of cyclopentane moieties of
brGDGTs, expressed as Cyclization of Branched Tetraethers (CBT), correlated
negatively with soil pH, while the number of methyl branches of brGDGTs, expressed
as Methylation of Branched Tetraethers (MBT), was dependent on annual mean air
temperature (MAT) and to a lesser extent on soil pH. The MBT/CBT proxies were
further corroborated by subsequent studies (e.g., Peterse et al., 2012; Sinninghe Damsté
et al., 2008; Yang et al., 2014a). Assuming that brGDGTs preserved in marine
sediments close to the Congo River outflow were derived from soils in the river
catchment, Weijers et al. (2007a) reconstructed large-scale continental temperature
changes in tropical Africa that span the past 25,000 years by using the MBT/CBT proxy.
More recently, De Jonge et al. (2013) used a tandem high performance liquid
chromatography-mass spectrometry (2D HPLC-MS) and identified a series of novel 6-
methyl brGDGTs which were previously coeluted with 5-methyl brGDGTs. This
finding resulted in the redefinition and recalibration of brGDGTs' indexes (e.g., De
Jonge et al., 2014; Xiao et al., 2015).
The premise of all brGDGT-based parameters is their source specificity, i.e.,
brGDGTs is only biosynthesized by bacteria thriving in soils and peat. Several studies,
however, observed different brGDGT compositions between marine sediments and
soils on adjacent lands, supporting in situ production of brGDGTs in marine
environments (e.g., Liu et al., 2014; Peterse et al., 2009a; Weijers et al., 2014; Zell et
al., 2014; Zhu et al., 2011), analogous to lacustrine settings (e.g., Sinninghe Damsté et
al., 2009; Tierney and Russell, 2009; Tierney et al., 2012) and rivers (e.g., De Jonge et
al., 2015; French et al., 2015a; Zell et al., 2015; Zhu et al., 2011). At the global scale,
Fietz et al. (2012) reported a significant correlation between concentrations of
brGDGTs and crenarchaeol ($p < 0.01$; $R^2 = 0.57–0.99$), suggesting that a common or
mixed source for brGDGTs and iGDGTs are actually commonplace in lacustrine and


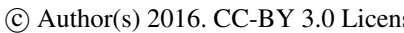


marine settings. More recently, Sinninghe Damsté (2016) examined tetraethers in
surface sediments from 43 stations in the Berau River delta (Kalimantan, Indonesia),
and their result, combined with data from other shelf systems, supported a widespread
biosynthesis of brGDGTs in shelf sediments especially at water depth of 50–300 m.

91        In continental shelf, river is the most important conduit for transporting brGDGTs

from land to sea because these compounds were either below the detection level
(Hopmans et al., 2004) or were present at low abundance (Fietz et al., 2013; Weijers et
al., 2014) in atmospheric dust. In the remote ocean where no direct impact from land
erosion via rivers takes place, eolian transport and in situ production became the most
important contributors to brGDGTs. Weijers et al. (2014) found that distributions of
African dust-derived brGDGTs were similar to those of soils but different from those
of distal marine sediments, providing a possibility to distinguish terrestrial vs. marine
brGDGTs based on their molecular compositions. Considering these facts, we attempt
to develop a robust index to assess the source of brGDGTs in marine environments. In
order to reach this objective, we first examined three cores in the Bohai Sea which are
subject to the Yellow River influence to different degree and compared the source
discerning capability of different brGDGT parameters. We then applied the most
sensitive parameter to globally distributed marine sediments and soils to test its validity.
Our study supplies an important step for improving accuracy of brGDGT-dervied
proxies and better understanding marine carbon cycle and paleoenvironments.

2 Material and methods
2.1  Study area and sampling

110       The Bohai Sea is a semi-enclosed shallow sea in northern China, extending about

550 km from north to south and about 350 km from east to west. Its area is 77,000 km$^2$
and means depth is 18 m (Hu et al., 2009). The Bohai Strait in the eastern portion is the
only passage connecting the Bohai Sea to the outer Yellow Sea. Several rivers,
including Yellow River, the second largest sediment-load river in the world, drain into
the Bohai Sea with a total annual runoff of $890 \times 10^8$ m$^3$. One gravity core with 64 cm





long (M1; 37.52°N, 119.32°E) was collected in July 2011, while other two cores were
collected in July 2013, namely M3 (38.66°N, 119.54°E; 53 cm long) and M7 (39.53°N,
120.46°E; 60 cm long), respectively (Fig. 2). The sites M1, M3 and M7 are located in
the south, the center and the north of the Bohai Sea, respectively. The cores were
transported to the lab where they were sectioned at 1 or 2 cm interval. The age model
was established on basis of $^{210}$Pb and $^{137}$Cs activity, showing that these cores cover the
sedimentation period of less than 100 years (Wu et al., 2013 and unpublished data).

2.2  Lipid extraction and analyses
The samples were freeze dried and homogenized with a mortar and pestle. After
the addition of $C_{46}$ GDGT (internal standard), the sediments (2–10 g) were
ultrasonically extracted with 25 ml dichloromethane(DCM)/methanol (3:1 v:v) for 15
min (3×). The combined extracts were concentrated by a rotary evaporator and
completely dried under a mild $N_2$ stream. The extracts were base hydrolyzed in 1 M
KOH/Methanol solution at 80 °C for 2 h. Neutral fractions were recovered by liquid-
liquid extraction with hexane, which were separated into two sub-fractions by 5 ml
hexane/DCM (9:1 v/v) and 5 ml DCM/Methanol(1:1 v/v), respectively, over silica gel
columns. The latter fraction containing GDGTs was filtered through 0.45 μm PTFE
filter before instrumental analyses.
The GDGTs were analyzed using an Agilent 1200 HPLC-atmospheric pressure
chemical ionization-triple quadruple mass spectrometry (HPLC-APCI-MS) system.
The polar fraction was dissolved in 300 μl hexane/EtOAc (84:16, v/v). Samples (10–
20 μl) were injected and the separation of 5- and 6-methyl brGDGTs was achieved with
two silica columns in sequence (150 mm×2.1 mm; 1.9 μm, Thermo Finnigan; USA) at
a constant flow of 0.2 ml/min. The solvent gradient was: 84% A (hexane) and 16% B
(EtOAc) for 5 min, increasing the amount of B from 16% at 5 min to 18% at 65 min,
and then to 100% B in 21 min. The column was flushed with 100% B for 4 min, and
then back to 84/16 A/B for 30 min in order to equilibrate the system. The APCI and MS
conditions were: vaporizer pressure of $4.2×10^5$ Pa, vaporizer temperature of 400 °C,
drying gas flow of 6 L min$^{-1}$, temperature of 200 °C, capillary voltage of 3500 V, and




corona current of 5 µA (3.2 kV). Samples were quantified based on comparisons of the
respective protonated-ion peak areas of each GDGT to the internal standard in selected
ion monitoring (SIM) mode. The protonated ions were m/z 1050, 1048, 1046, 1036,
1034, 1032, 1022, 1020, 1018 for brGDGTs, 1302, 1300, 1298, 1296, 1292 for iGDGTs
and 744 for $C_{46}$ GDGT.

2.3  Parameter calculation and statistics
The BIT, MBT, Methyl Index (MI), Degree of Cyclization (DC) of brGDGTs and
weighted average number of cyclopentane moieties for tetramethylated brGDGTs
(#Rings$_{tetra}$) were calculated according to the definitions of Hopmans et al. (2004),
Weijers et al. (2007b), Zhang et al. (2011), Sinninghe Damsté et al. (2009) and
Sinninghe Damsté (2016), respectively.
$$\text{BIT} = \frac{\text{Ia} + \text{IIa} + \text{IIIa}}{\text{Ia} + \text{IIa} + \text{IIIa} + \text{IV}} \quad (1)$$
$$\text{MBT} = \frac{\text{Ia} + \text{Ib} + \text{Ic}}{\text{Ia} + \text{IIa} + \text{IIIa} + \text{Ib} + \text{IIb} + \text{IIIb} + \text{Ic} + \text{IIc} + \text{IIIc}} \quad (2)$$
$$\text{MI} = 4 \times (\text{Ia} + \text{Ib} + \text{Ic}) + 5 \times (\text{IIa} + \text{IIb} + \text{IIb}) + 6 \times (\text{IIIa} + \text{IIIb} + \text{IIIc}) \quad (3)$$
$$\text{DC} = \frac{\text{Ib} + \text{IIb}}{\text{Ia} + \text{IIa} + \text{Ib} + \text{IIb}} \quad (4)$$
$$\text{#Rings}_{tetra} = \frac{\text{Ib} + 2 * \text{Ic}}{\text{Ia} + \text{Ib} + \text{Ic}} \quad (5)$$
where roman numbers denote relative abundance of compounds depicted in Fig. 1. In
this study, we used two silica LC columns in tandem and successfully separated 5- and
6-methyl brGDGTs. However, many previous studies (e.g., Weijers et al., 2006) used
one LC column and did not separate 5- and 6-methyl brGDGTs. Considering this, we
combined 5-methyl and 6-methyl brGDGT as one compound in this study, for example,
IIIa denotes the total abundance of brGDGT IIIa and IIIa' in figure 1.
An analysis of variance (ANOVA) was conducted for different types of samples
to determine if they differ significantly from each other. The SPSS 16.0 software
package (IBM, USA) was used for the statistical analysis. Squared Pearson correlation
coefficients ($R^2$) reported have an associated $p$ value < 0.05.





2.4  Data compilation of global soils and marine sediments

175        The dataset in this study are composed of GDGTs from 1354 globally distributed

soils and 589 marine sediments (Fig. 2). These samples span a wide area from 75.00ºS
to 79.28ºN and 168.08ºW to 174.40ºE and have water depth of 1.0 to 5521 m. The
marine samples are from the South China Sea (Dong et al., 2015; Hu et al., 2012; Jia et
al., 2012; O'Brien et al., 2014), Caribbean Sea (O'Brien et al., 2014), western equatorial
Pacific Ocean (O'Brien et al., 2014), southeast Pacific Ocean (Kaiser et al., 2015), the
Chukchi and Alaskan Beaufort Seas (Belicka and Harvey, 2009), eastern Indian Ocean
(Chen et al., 2014), East Siberian Arctic Shelf (Sparkes et al., 2015), Kara Sea (De
Jonge et al., 2016; De Jonge et al., 2015), Svalbard fjord (Peterse et al., 2009a), Red
Sea (Trommer et al., 2009), the southern Adriatic Sea (Leider et al., 2010), Columbia
estuary (French et al., 2015b), globally distributed distal marine sediments (Weijers et
al., 2014) and the Bohai Sea (this study). Soil samples are from the Svalbard (Peterse
et al., 2009b), Columbia (French et al., 2015b), China (Ding et al., 2015; Hu et al., 2016;
Xiao et al., 2015; Yang et al., 2013; Yang et al., 2014a; Yang et al., 2014b), globally
distributed soils (De Jonge et al., 2014; Peterse et al., 2012; Weijers et al., 2006),
California geothermal (Peterse et al., 2009b), France and Brazil (Huguet et al., 2010),
western Uganda (Loomis et al., 2011), the USA (Tierney et al., 2012), Tanzania
(Coffinet et al., 2014), Indonesian, Vietnamese, Philippine, China and Italia (Mueller-
Niggemann et al., 2016).

3 Results and discussion
3.1  Distribution and source of brGDGTs in Bohai Sea

197        Both iGDGTs including crenarchaea and brGDGTs were detected in Bohai Sea

sediments. For brGDGTs, a total of 15 compounds were identified including three
tetramethylated brGDGTs (Ia, Ib and Ic), six pentamethylated brGDGTs (IIa, IIb, IIc,
IIa', IIb' and IIc') and six hexamethylated brGDGTs (IIIa, IIIb, IIIc, IIIa', IIIb' and
IIIc'). In order to evaluate provenances of brGDGTs, we calculated various parameters
including the BIT index, percentages of tetra-, penta- and hexa-methylated brGDGTs,
#rings for tetramethylated brGDGTs, DC, MI, MBT, brGDGTs IIIa/IIa and Ia/IIa (Table





1). The values of the BIT index ranged from 0.27 to 0.76 in the core M1, which are much higher than that in the core M3 (0.04–0.25) and the core M7 (0.04–0.18). Such difference is expectable since the site M1 is closest to the Yellow River outflow, and receives more terrestrial organic carbon than other two sites (Fig. 2). However, the BIT index itself has no ability to distinguish terrestrial vs. aquatic brGDGTs because brGDGTs and crenarchaea used in this index are thought to be specific for soil organic carbon and marine organic carbon, respectively (Hopmans et al., 2004). For individual brGDGTs, the core M1 is characterized by significantly higher percentage of brGDGT IIa (28±1%) than the core M2 (18±1%) and the core M3 (18±0%; Fig. 3). We performed ANOVA for a variety of brGDGTs' parameters, and the results (Table 1) show that all parameters except MI can distinguish Chinese soils from Bohai Sea sediments, but only the IIIa/IIa ratio can completely separate Chinese soils (0.39±0.25; Mean±SD; same hereafter), M1 sediments (0.63±0.06), M3 sediments (1.16±0.12) and M7 sediments (0.93±0.07) into four groups.

Three factors may account for the occurrence of higher IIIa/IIa ratio in the Bohai Sea sediments than Chinese soils: selective degradation during land to sea transport, admixture of river produced brGDGTs and in situ production of brGDGTs in sea. Huguet et al. (2008; 2009) reported that iGDGTs (i.e., crenarchaea) was degraded at a rate of 2-fold higher than soil derived brGDGTs under long term oxygen exposure in the Madeira Abyssal Plain, leading to increase of the BIT index. Such selective degradation, however, cannot explain significant different IIIa/IIa ratio between the Chinese soils and Bohai Sea sediments because unlike crenarchaea, both IIIa and IIa belong to brGDGTs with similar chemical structures and thus have similar degradation rates. In situ production of brGDGTs in rivers is a widespread phenomenon, and can change brGDGT compositions in sea when they were transported there (e.g., De Jonge et al., 2015; Zell et al., 2015; Zhu et al., 2011). However, this effect is minor in the Yellow River because extremely high turbidity (up to 220 kg/m3 during the flood season; Ren and Shi, 1986) greatly constrain the growth of aquatic organisms. The studies along lower Yellow River-estuary-coast transect suggested that brGDGTs in surface sediments were primarily a land origin (Wu et al., 2014). Therefore, the





enhanced IIIa/IIa values in the Bohai Sea sediments is caused by in situ production of
brGDGTs. An increasing trend from the site M1 (0.63±0.06) to M7 (0.93±0.07) then to
M3 (1.16±0.12) reflects variability in relative contribution of autochthonous (lower
IIIa/IIa) and allochthonous (higher IIIa/IIa) brGDGTs. The site M1 is adjacent to the
Yellow River mouth and receives the largest amount of terrestrial organic matter,
causing lower IIIa/IIa values. In contrast, the site M3 located in central Bohai Sea
comprises of the least amount of terrestrial organic matter, resulting in higher IIIa/IIa
values. The intermediate IIIa/IIa values at the site M7 is attributed to moderate land
erosion nearby northern Bohai Sea (Fig. 2). Such distribution pattern strongly suggests
that the IIIa/IIa ratio is a sensitive indicator for assessing source of brGDGTs in the
Bohai Sea.

3.2 Regional and global validation of brGDGT IIIa/IIa
To test whether the IIIa/IIa ratio is valid in other environments, we apply it to the
Svalbard (Peterse et al., 2009a), the Yenisei River outflow (De Jonge et al., 2015) and
the East Siberian Arctic Shelf (Sparkes et al., 2015). By comparing the compositions of
brGDGTs in Svalbard soils and nearby fjord sediments, Peterse et al. (2009a) indicated
that sedimentary organic matter in fjords was predominantly a marine origin. A plot of
BIT vs. IIIa/IIa (Fig. 4a) clearly grouped the samples into two groups which correspond
to soils (>0.75 for BIT and <1.0 for IIIa/IIa) and marine sediments (<0.3 for BIT
and >1.0 for IIIa/IIa). Another line of evidence is from De Jonge et al. (2015) who
examined brGDGTs in core lipids (CLs) and intact polar lipids (IPLs) in the Yenisei
River outflow. As the IPLs are rapidly degraded in the environment, they can be used
to trace living or recently living material, while the CLs are generated via degradation
of the IPLs after cell death (Lipp et al., 2008; White et al., 1979). The compiling of
brGDGTs from De Jonge et al. (2015) shows significant difference of the IIIa/IIa ratio
between the IPL fractions (>1.0) and CL fractions (<0.8; Fig. 4b). Such disparity
supports that brGDGTs produced in marine environments have higher IIIa/IIa values
because labile intact polar brGDGTs are mainly produced in situ, whereas recalcitrant
core brGDGTs are composed of more allochthonous terrestrial components. Sparkes et





al. (2015) examined brGDGTs in surface sediments across the East Siberian Arctic
Shelf (ESAS) including the Dmitry-Laptev Strait, Buor-Khaya Bay, ESAS nearshore
and ESAS offshore. The plot of BIT vs. IIIa/IIa again results into two groups, one group
with lower BIT values (<0.3) and higher IIIa/IIa values (0.8–2.3) mainly from ESAS
offshore, and another group with higher BIT values (0.3–1.0) and lower IIIa/IIa values
(0.4–0.9) from the Dmitry-Laptev Strait, Buor-Khaya Bay and ESAS nearshore (Fig.
4c). A strong linear correlation was observed between the IIIa/IIa ratio and the distance
from river mouth ($R^2$=0.58; $p<0.05$; Fig. 4d), in accord with the data of the BIT index
and $\delta^{13}C_{org}$ (Sparkes et al., 2015). All lines of evidence support that marine-derived
brGDGTs have higher IIIa/IIa values than terrestrial derived brGDGTs.
We further compile all available data in literatures representing globally
distributed soils and marine sediments (Fig. 5). The statistical analysis clearly showed
that at the global scale, the IIIa/IIa ratio was significantly higher in marine sediments
than soils ($p<0.05$). An exception was observed for Red Sea sediments which have
unusually low IIIa/IIa values (0.39±0.21). The Red Sea has a restricted connection to
the Indian Ocean via the Bab el Mandeb. This, combined with high insolation, litter
precipitation and strong winds result in surface water salinity up to 41 in the south and
36 in the north of the Red Sea (Sofianos et al., 2002). Under such extreme environment,
distinct populations of Crenarchaeota may be developed and produced GDGTs different
from that in other marine settings (Trommer et al., 2009).
Overall, the global distribution of IIIa/IIa presents the highest level in many deep
sea sediments (2.6~5.1), the lowest level in soils (<1.0), and an intermediate level in
sediments from bays, coastal areas or marginal seas (0.87~2.62; Fig. 5). These results
are consistent with our data from the Bohai Sea, and confirm that the IIIa/IIa ratio is a
useful proxy for tracing the source of brGDGTs in marine sediments at regional and
global scales.
Why do soils have lower IIIa/IIa values than marine sediments? It is well known
that relative number of methyl groups (e.g., MBT) has a negative correlation with soil
pH and a positive correlation with MAT (Peterse et al., 2012; Weijers et al., 2007b).
The IIIa/IIa ratio is actually an abundance ratio of hexamethylated to pentamethylated





brGDGT, and thus may be also controlled by ambient temperature and pH of source
organisms. Unlike iGDGTs which is well known to be mainly produced by
Thaumarchaeota (Schouten et al., 2008; Sinninghe Damsté et al., 2002), the marine
source of brGDGTs remains elusive. Here, we assume that marine organisms producing
brGDGTs response to ambient temperature in a same way as soil bacteria producing
brGDGTs, i.e., a negative correlation between relative number of methyl group of
brGDGTs and ambient temperature. However, even if this hypothesis is tenable,
temperature is still unable to explain observed distribution patterns of the IIIa/IIa ratio
because both soils and marine sediments are globally distributed and their temperatures
(MAT vs. sea surface temperature) have no systematic difference. Alternatively, the
analysis of global soil data of Peterse et al. (2012) shows that the IIIa/IIa ratio has a
positive correlation with soil pH ($R^2$=0.43). In this study, the majority of soils are acidic
or neutral (pH<7.3) and only 8% of soils have pH of >8.0 except for those from semi-
arid and arid regions (e.g., Yang et al., 2014a), whereas seawater is constantly alkaline
with pH of 8.2 on average. With this systematic difference, bacteria living in soils tend
to produce higher proportions of brGDGT IIIa, whereas unknown marine organisms
tend to biosynthesize higher proportions of brGDGT IIa if they response to ambient pH
in a similar way as soil bacteria in term of biosynthesis of brGDGTs (Peterse et al.,

312  2012).


3.3  Implication of IIIa/IIa on other brGDGT proxies
Because brGDGTs can be produced in marine settings, they are no longer specific
for soil organic matter, which inevitably affects brGDGT proxies (e.g., BIT, MBT/CBT).
The plot of BIT vs. IIIa/IIa on basis of global dataset shows that the IIIa/IIa ratio has
the value of <0.59 for 90% of soil samples and >0.92 for 90% of marine sediments (Fig.
6). Considering this fact, we propose that the IIIa/IIa ratio of <0.59 and >0.92 represents
terrestrial (or soil) and marine endmembers, respectively. The BIT index has the value
of >0.67 for 90% of soils and <0.16 for 90% of marine sediments (Fig. 6). Overall, the
BIT index decreased with increasing IIIa/II values($BIT = 1.08 \times 0.28^{\frac{IIIa}{IIa}} - 0.03$; $R^2 =$





0.77; Fig. 6), suggesting that both the IIIa/IIa and BIT are useful indexes for assessing
soil organic carbon in marine settings. However, when the BIT index has an
intermediate value (i.e., 0.16 to 0.67), it is not valid to determine the provenance of
brGDGTs. For example, several marine samples having BIT values of ~0.35 show a
large range of IIIa/IIa (0.4 to 2.4; Fig. 6), suggesting that the source of brGDGTs can
vary case by case. Under this situation, the measurement of the IIIa/IIa ratio is strongly
recommended.
The different IIIa/IIa values between land and marine endmembers may supply an
approach to quantify the contribution of soil organic carbon in marine sediments.
Similar to the BIT index, we used a binary mixing model to calculate percentage of soil
organic carbon (%$OC_{soil}$) as follow:
$$\%OC_{soil} = \left[ \frac{[IIIa/IIa]_{sample} - [IIIa/IIa]_{marine}}{[IIIa/IIa]_{soil} - [IIIa/IIa]_{marine}} \right] * 100 \quad (6)$$
Where $[IIIa/IIa]_{sample}$, $[IIIa/IIa]_{soil}$ and $[IIIa/IIa]_{marine}$ are the abundance ratio of brGDGT
IIIa/IIa for samples, soils and marine sediments devoid of terrestrial influences,
respectively.
We applied this binary mixing model to the East Siberian Arctic Shelf because the
data of BIT, $\delta^{13}C_{org}$ and distance from river mouth are all available (Sparkes et al., 2015).
With the distance from river mouth increasing from 25 to >700 km, the BIT, IIIa/IIa
and $\delta^{13}C_{org}$ change from 0.95 to 0, 0.53 to 2.21 and –27.4‰ to –21.2‰, respectively,
reflecting spatial variability of sedimentary organic carbon sources. For the BIT index,
we used 0.97 and 0.01 as terrestrial and marine endmember values based on previous
studies for Arctic surrounding regions (De Jonge et al., 2014; Peterse et al., 2014),
which are similar to global average values (Hopmans et al., 2004). For $\delta^{13}C_{org}$, we chose
–27‰ and –20‰ as C3 terrestrial and marine organic carbon endmembers (Meyers,
1997 and references therein). For the IIIa/IIa ratio, we used a global average value of
marine sediments (1.6) and soils (0.24), respectively, based on this study. By applying
these endmember values into Eq. 6, we calculated percentage of soil organic carbon
(%$OC_{soil}$). We removed a few data points if their calculated %$OC_{soil}$ were greater than
100% or below 0%. It should be noted that the endmember value will affect quantitative





results, but does not change a general trend of %OC$_{soil}$. The results based on all three
parameters show a decreasing trend seawards (Fig. 7). However, the %OC$_{soil}$ based on
$\delta^{13}C_{org}$ is the highest (75±18%), followed by that from the IIIa/IIa ratio (58±15%) and
then that from the BIT index (43±27%). This difference have been explained by that
$\delta^{13}C_{org}$ is a bulk proxy for marine vs. terrestrial influence of sedimentary organic carbon
(SOC), whereas the BIT index is for a portion of the bulk SOC, i.e., soil OC (Walsh et
al., 2008) or fluvial OC (Sparkes et al., 2015). For the estimated %OC$_{soil}$, $\delta^{13}C_{org}$
presents a stronger positive correlation with the IIIa/IIa ratio (R$^2$=0.49) than the BIT
index (R$^2$=0.45), suggesting that the IIIa/IIa ratio may serve a better proxy for
quantifying soil organic carbon than the BIT index because it is less affected by
selective degradation of branched vs. isoprenoid GDGTs and high production of
crenarchaea in marine environments (Smith et al., 2012).

4 Conclusions
Based on a detailed study on GDGTs for three cores in the Bohai Sea and
compiling of GDGT data from globally distributed soils and marine sediments, we have
reached several important conclusions. Firstly, the ratio of brGDGTs IIIa/IIa is
generally lower than 0.59 in soils, but higher than 0.92 in marine sediments devoid of
significant terrestrial inputs, making it a sensitive proxy for assessing soil vs. marine
derived brGDGTs at regional and global scales. Secondly, in situ production of
brGDGTs in marine environments is a ubiquitous phenomenon, which is particularly
important for those marine sediments with low BIT index (<0.16) where brGDGTs are
exclusively of a marine origin. Thirdly, a systemic difference of the IIIa/IIa value
between soils and marine sediments reflects an influence of pH rather than temperature
on the biosynthesis of brGDGTs by source organisms. Given these facts, we strongly
recommend to calculate the IIIa/IIa ratio before estimating organic carbon source,
paleo-soil pH and MAT based on the BIT and MBT/CBT proxies.

Acknowledgements. The work was financially supported by the National Science
Foundation of China (41476062). We are grateful for X. Dang for GDGT analyses. G.





Jia, J. Hu, A. Leider, G. Mollenhauer, G. Trommer and R. Smith are thanked for kindly
supplying GDGT data.

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



Fig.1. Chemical structures of branched GDGTs and crenarchaeol.

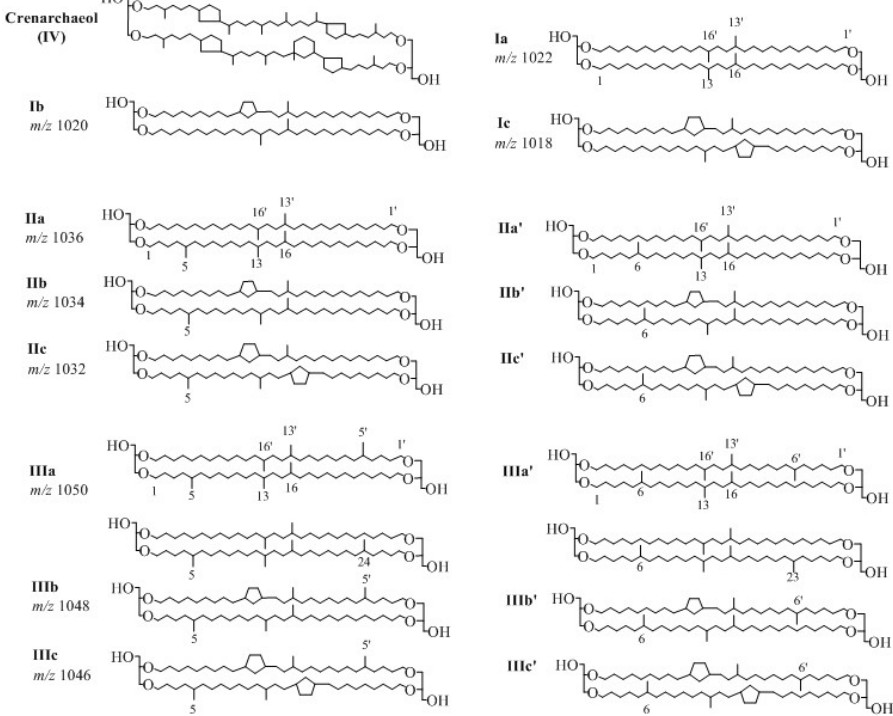




















Fig.2. Location of the samples used in this study. White circles and black circles
indicate the soils and marine sediments, respectively. Red crosses denote three sediment
cores (M1, M3 and M7) in the Bohai Sea. YR is the Yellow River.

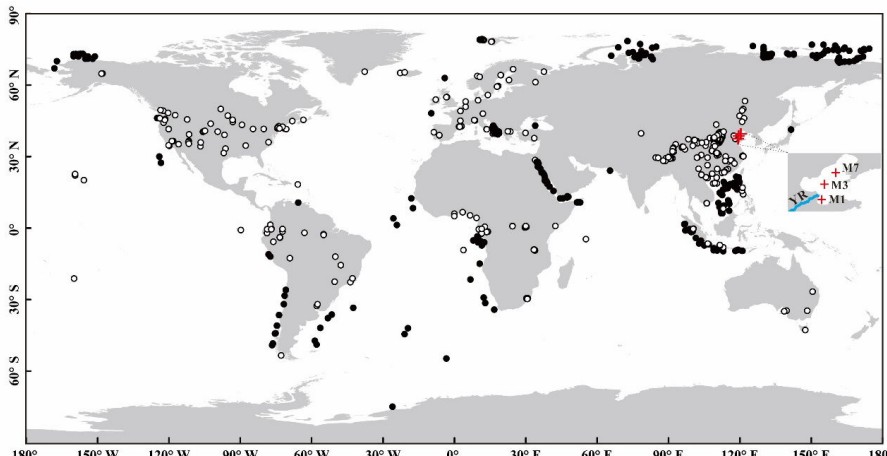























Fig.3. Averaged percentages of individual brGDGTs in soils (a), core M1 (b), M3 (c)
and M7 (d). The soil data are from Yang et al. (2014a).

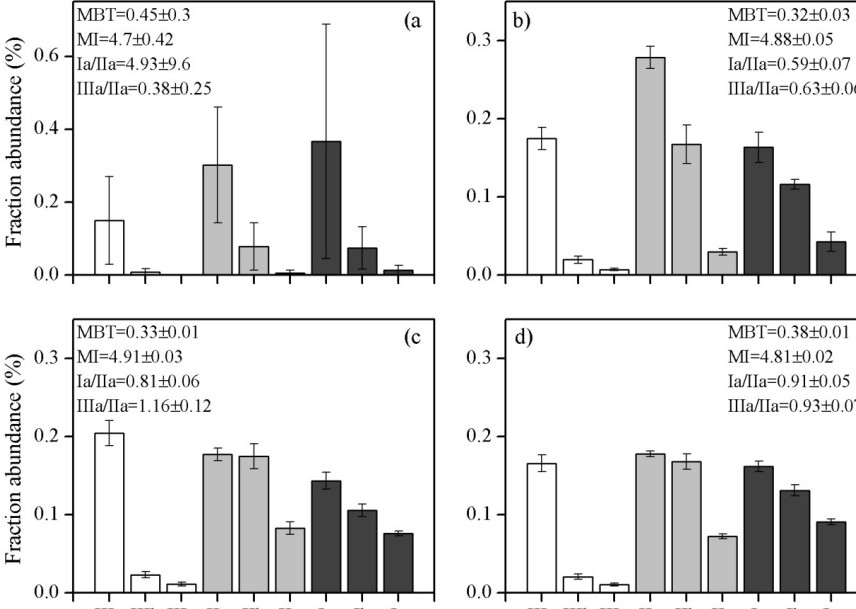




















Fig. 4. a) The relationship between brGDGT IIIa/IIa ratio and the BIT index of samples
from Peterse et al. (2009a); b) histograms of brGDGT IIIa/IIa ratio of the core lipids
(CLs) and intact polar lipids (IPLs) in samples from De Jonge et al. (2015); c) the
relationship between brGDGT IIIa/IIa ratio and the BIT index in samples from Sparkes
et al. (2015); d) the relationship between brGDGT IIIa/IIa ratio and distance from river
mouth in samples from Sparkes et al.(2015).

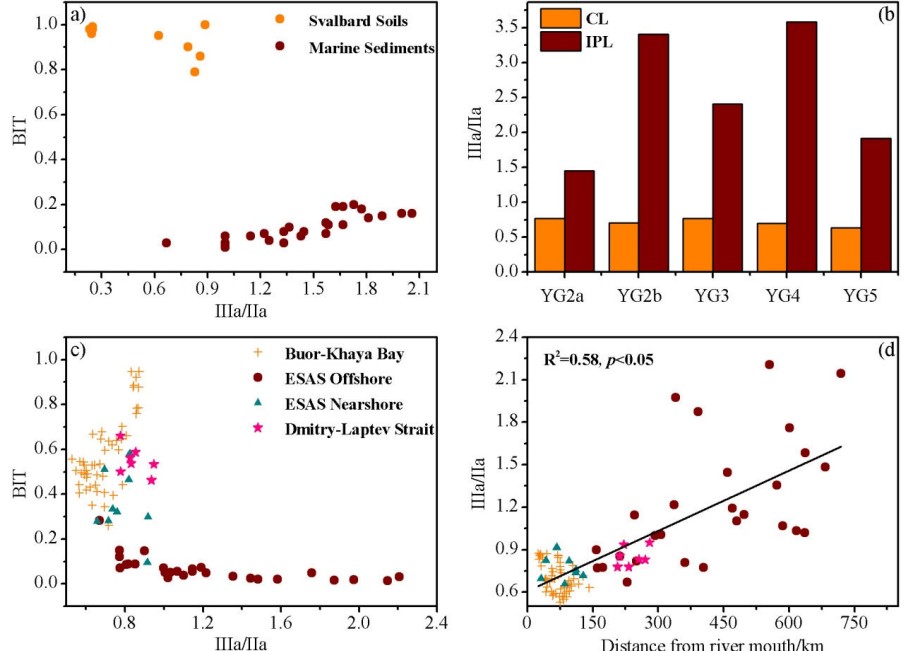
















Fig. 5. Global distribution pattern of brGDGT IIIa/IIa ratio in soils and marine
sediments.

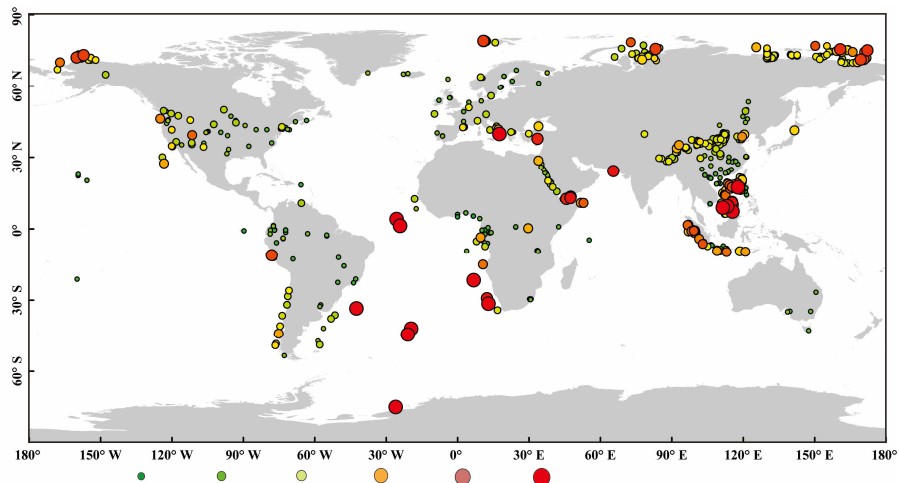



Fig. 6. Relationship between the IIIa/IIa ratio and the BIT index of globally distributed
samples: soils (orange circle) and marine sediments (red circle). Dashed lines represent
lower or upper threshold values for 90% of soils/sediments.

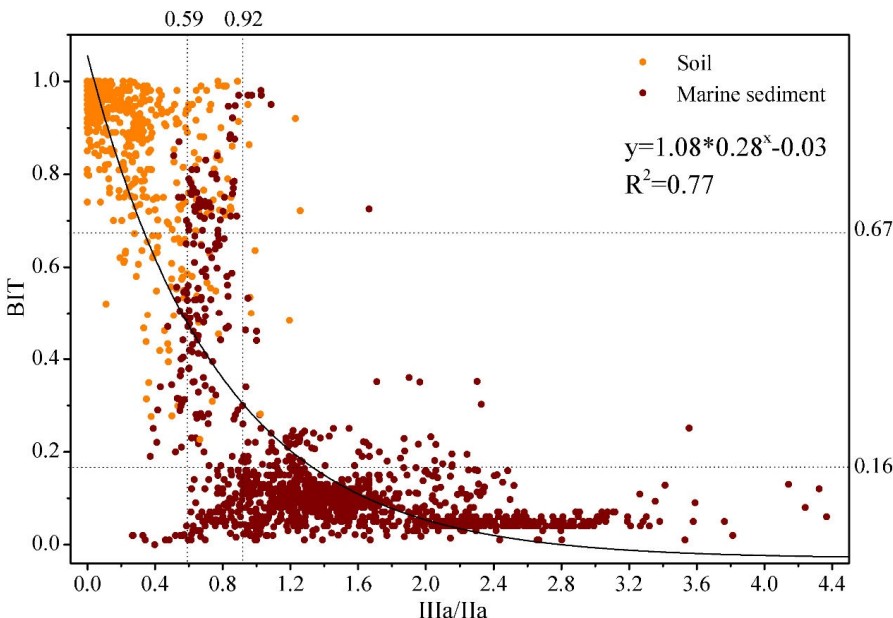




















Fig. 7. Percentage of soil organic carbon (%OC$_{soil}$) or terrestrial organic carbon
(%OC$_{terr}$) based on a binary mixing model of BIT (a), $\delta^{13}$C$_{org}$ (b) and IIIa/IIa (c) for the
East Siberian Arctic Shelf (Sparkes et al., 2015).

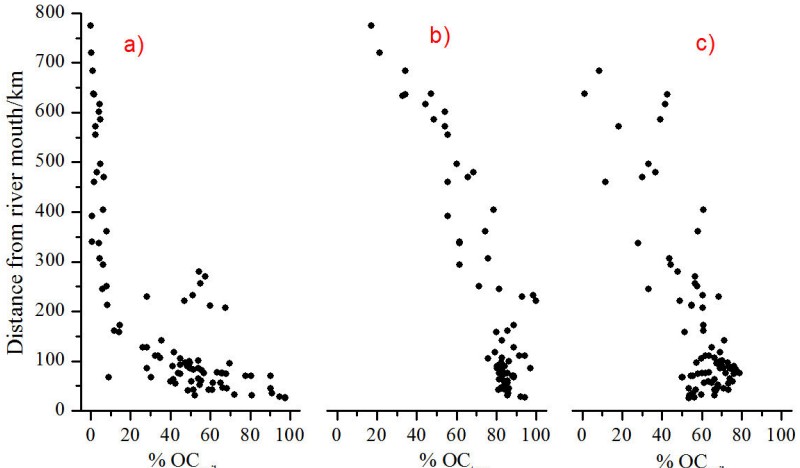





















Table 1: Parameters including brGDGTs IIIa/IIa, Ia/IIa, the BIT index, MBT, MI, DC,
percentages of tetra-, penta- and hexa-methylated brGDGTs, and the weighted average
number of cyclopentane moieties (#rings for tetramethylated brGDGTs) based on the
GDGTs from three cores (M1, M3 and M7) in the Bohai Sea. Different letters (a, b, c,
d) represent significant difference at the level of $p < 0.05$.

| Indexes | Soil | M1 | M3 | M7 |
|---|---|---|---|---|
| IIIa/IIa | 0.39±0.25 (a) | 0.63±0.06 (b) | 1.16±0.12 (c) | 0.93±0.07 (d) |
| Ia/IIa | 4.93±9.60 (a) | 0.59±0.07 (b) | 0.81±0.06 (b) | 0.91±0.05 (b) |
| BIT | 0.75±0.22 (a) | 0.50±0.19 (b) | 0.14±0.06 (c) | 0.11±0.03 (c) |
| MBT | 0.45±0.30 (a) | 0.32±0.03 (b) | 0.33±0.01 (b) | 0.38±0.01 (ab) |
| MI | 4.70±0.42 (a) | 4.88±0.05 (b) | 4.91±0.03 (b) | 4.81±0.02 (ab) |
| DC | 0.31±0.21 (a) | 0.62±0.03 (b) | 0.79±0.03 (c) | 0.82±0.02 (c) |
| %tetra | 0.45±0.30 (a) | 0.32±0.03 (b) | 0.33±0.01 (c) | 0.38±0.01 (c) |
| %hexa | 0.16±0.12 (a) | 0.20±0.02 (b) | 0.24±0.02 (b) | 0.20±0.01 (b) |
| %penta | 0.39±0.20 (a) | 0.48±0.02 (b) | 0.44±0.02 (b) | 0.42±0.01 (b) |
| #Rings$_{tera}$ | 0.20±0.15 (a) | 0.39±0.03 (b) | 0.47±0.02 (c) | 0.47±0.02 (c) |

