# Peer review of "Ubiquitous production of branched glycerol dialkyl glycerol tetraethers (brGDGTs) in"

_Biogeosciences, 2016_

## Referee Comment (RC1) · Anonymous Referee #1 · 27 Jun 2016

Review of the manuscript "Ubiquitous production of branched glycerol dialkyl glycerol tetraethers (brGDGTs) in global marine environments: a new source indicator for brGDGTs", by Xiao and co-authors.

The authors present an interesting manuscript, based on an extensive dataset compiled from previous publications. The figures are well chosen and convey the message clearly. The IIIa/IIa ratio can be used to identify those sediments that are characterized by a dominant marine production of brGDGTs. This ratio can complement the BIT index, and has the added feature that the degradation of marine-sourced IIIa vs IIa is

expected to happen to the same extent (while crenarchaeol has a more different chemical structure and possibly a different source organism and thus a different degradation potential).

The implication and possible use this ratio has for tracing soil OM transport is exemplified and convincing. However, for palaeoenvironmental reconstructions, there is no guideline on how to deal with sediments with 'mixed signals'. Here, discussing the compounds separately can help, as it will point the authors towards which proxies can and can't be used in the case of in-situ production of brGDGTs

Main comments (short, they are elaborated on below).

It is interesting that the IIIa/IIa ratio increase significantly in offshore marine sediments. However, the authors have not attempted to explain this by comparing the compounds this ratio is composed of (IIIa, IIIa', IIa or IIa'). This lessens the value of this study, by narrowing its implication for palaeoenvironmental studies. In the Kara Sea (Arctic Ocean) De Jonge et al. (2016) have clearly shown that brGDGT IIIa' increases in increasingly marine conditions (Yenisei River outflow), while brGDGT IIa' does not. The Iberian Sea (Sinninghe Damste et al., shows a different pattern). If the authors can shed light on which mechanism acts on marine sediments globally, this has implications for which temperature proxies can be used (also see De Jonge et al. (2016)).

I find the reasoning behind the absence of a temperature difference between soil/peat and marine brGDGTs incomplete. I expect a very large difference in temperature between soil and marine bottom water, even at similar latitudes.

The introduction of previous studies describing marine in-situ produced brGDGTs is too concise. Furthermore, in the discussion I miss how the conclusions from the authors fit with previously published manuscripts? Can we say anything about the water depths at which brGDGTs are produced?

Minor comments:

L14. Rephrase this so the "presumed source" of brGDGTs (soil, peat) is introduced first.

L33: Use 'have been' instead of 'have become'.

L43: Rephrase, this is a confusing sentence. The stereoconfiguration of the glycerol moiety indicates that they are produced by bacteria, not the fact that they are abundant in soils.

L49: Please include that 15 brGDGT compounds are generally encountered in soils. Besides the variation in the number of methyl groups and cyclopentane moieties, the location of the outer branches has been shown to shift as well.

L 54: Use Thaumarchaeota instead of crenarchaea.

L 58: Here, you can also refer to 'Weijers et al. (2014), Constraints on the sources of branched tetraether membrane lipids in distal marine sediments, OG 72'.

L 87-90. As this manuscript discusses brGDGTs produced in marine systems, I would expand a bit more on all studies that have provided evidence for the in-situ production of brGDGTs in the marine environment (instead of just listing them up). Now, only the recent Sinninghe Damste (2016) paper is introduced.

L 91-99 could be restructured, they are not easy to understand.

L 106: 'the marine carbon cycle'

L 112. 'the mean depth is' and 'the Bohai Strait, at the east'.

L 114: 'the second largest river in the world, concerning sediment load (+reference)'

L 115: 'One gravity core of 64 cm was...'

L 118: respectively can be removed here.

L 125: If this extraction and separation protocol has been described elsewhere, you can simply refer to this original publication. The same goes for the analysis of the

GDGTs on the LC system.

L 183: This can be rewritten as: De Jonge et al., 2015, 2016.

L 206: Can the authors not give an indication at which BIT values (both on the local and global scale) the proportion of marine brGDGT becomes problematic? This would be useful from the viewpoint of palaeoclimate reconstructions.

L 235-237: Is a repetition of the L 237-242.

L 282.    It surprises me that the authors indicate here that Crenarchaeota/Thaumarchaeota are the probable producers of marine brGDGTs. Is there any indication that this would be the case? Alternatively, I would remove this statement.

L 303. The argument that continental and marine temperatures are significantly different is put aside much too quickly. Indeed, they are both globally distributed, but the temperature of your water bodies will be much more stable throughout the year (which has an implication of the production temp as soil-derived brGDGTs are thought to be produced mainly in spring-autumn, especially at sites that are partially frozen throughout the year. Furthermore, if marine brGDGTs are produced at the sediment/water interface, this will of course be much colder than the sea surface temperature.

Taking this into account, I doubt that the authors will be able to make a strong case on their proposed absence of a temperature difference between soil and marine brGDGTs.

L 304. If the authors want to discuss this trend between soil pH and III/II, they have to provide a plot. Does this trend also apply for more extreme pH values? Can it be strengthened by determining which compound causes this trend (IIa, IIa', IIIa, IIIa')?

L 308: The pH of marine water is indeed fairly stable, but it can be very different in pore waters in the sediments. This should be mentioned.

L 367: 'and a compilation of'
[Figure]

L 364-367: I do not agree that the authors have enough evidence and data on this to make this conclusion.

References: please check the manuscripts guidelines. Journal names are to be abbreviated.

General: In the manuscript text, the authors should pay attention to the order of references. Older references should come first.

---

## Author Comment (AC1) · 17 Jul 2016

Response to reviewer 1's comments: On behalf of my coauthors, I really appreciate the reviewer to acknowledge the merit of our work. As the reviewer said, "The authors present an interesting manuscript, based on an extensive dataset compiled from previous publications. The figures are well chosen and convey the message clearly." In addition, we also thank the reviewer providing a number of useful comments, which are helpful to improve our manuscript. Here, we tried our best to address the reviewer's comments point by point.

[Figure]

Anonymous Referee #1 1) Main comments (short, they are elaborated on below). It is interesting that the IIIa/IIa ratio increase significantly in offshore marine sediments. However, the authors have not attempted to explain this by comparing the compounds this ratio is composed of (IIIa, IIIa', IIa or IIa'). This lessens the value of this study, by narrowing its implication for palaeoenvironmental studies. In the Kara Sea (Arctic Ocean), De Jonge et al. (2016) have clearly shown that brGDGT IIIa' increases in increasingly marine conditions (Yenisei River outflow), while brGDGT IIa' does not. The Iberian Sea (Sinninghe Damste et al., shows a different pattern). If the authors can shed light on which mechanism acts on marine sediments globally, this has implications for which temperature proxies can be used (also see De Jonge et al. (2016)).

Response: In our study, we used 2D LC-MS to separate 5-methy and 6-methyl brGDGTs. The reason we combined them together in the manuscript is that most previous studies using one dimensional LC-MS did not separate these two types of isomers. The first study to report 6-methyl brGDGTs was published in 2013 by De Jonge et al. So far only very limited studies paid attention to this issue. Nevertheless, we agree with the reviewer that the separation of 5-methyl and 6-methyl brGDGTs may provide more accurate proxies for source and environmental information of brGDGTs. In the revised manuscript, we added comment on this points We wrote in the conclusion as "We also note a relatively large scatter of the IIIa/IIa ratio within both terrestrial and marine realms, and different environmental responses of 5-methyl and 6-methyl brGDGTs (e.g., De Jonge et al., 2014, 2016; Xiao et al., 2015). As a result, the separation of these two types of isomers is needed in future studies to develop more accurate brGDGTs' proxies." We expect more data about 5-methyl and 6-methyl brGDGTs available in future, so we can compile them and develop new molecular proxies.

2) I find the reasoning behind the absence of a temperature difference between soil/peat and marine brGDGTs incomplete. I expect a very large difference in temperature between soil and marine bottom water, even at similar latitudes.

Response: This is a good comment. Recent studies suggested that the production

of GDGTs in deep water is possible. If so, large temperature gradient between surface and deep water in ocean inevitably affects brGDGTs' compositions. In the revised manuscript, we consider this factor for different brGDGTs' compositions between land and sea. In section 3.2, we rewrote the whole paragraph (line 486-549) as "Why do marine sediments have higher IIIa/IIa values than soils? It has been reported that relative number of methyl groups positively correlates with soil pH and negatively correlates with MAT (Peterse et al., 2012; Weijers et al., 2007b). The IIIa/IIa ratio is actually an abundance ratio of hexamethylated to pentamethylated brGDGT, and thus is also affected by ambient temperature and pH. Unlike iGDGTs which is well known to be mainly produced by Thaumarchaeota (Schouten et al., 2008; Sinninghe Damsté et al., 2002), the marine source of brGDGTs remains elusive. Here, we assume that marine organisms producing brGDGTs response to ambient temperature in the same way as those soil bacteria producing brGDGTs, i.e., a negative correlation between relative number of methyl group of brGDGTs and ambient temperature. In order to evaluate temperature effect on brGDGTs' compositions, we need consider the locale where brGDGTs are produced. If brGDGTs in marine environments are predominantly produced in euphotic zone, a significant difference for the IIIa/IIa ratio would not be observed between soils and marine sediments because both soils and marine sediments are globally distributed, leading to no systematic difference between soil temperature and sea surface temperature. Alternatively, if brGDGTs in marine sediments are partially derived from deep-water dwelling or benthic organisms, cold deep water (generally 1‒2 oC) would cause higher IIIa/IIa values in marine sediments, as we observed. Besides temperature, pH can also alter compositions of brGDGTs (Weijers et al., 2007). Based on global soil data, the IIIa/IIa ratio shows a strong positive correlation with soil pH ($R2=0.51$; Fig. 6). In our study, the majority of soils are acidic or neutral (pH<7.3) and only 8% of soil samples mainly from semi-arid and arid regions have pH of >8.0 (e.g., Yang et al., 2014a). In contrast, seawater is constantly alkaline with a mean pH of 8.2. With this systematic difference, bacteria living in soils tend to produce higher proportions of brGDGT IIa, whereas unknown marine organisms tend

to biosynthesize higher proportions of brGDGT IIIa if they response to ambient pH in a similar way as soil bacteria in term of biosynthesis of brGDGTs. Taking together, we attributed the occurrence of higher IIIa/IIa values in marine sediments to higher pH and lower deep water temperature. Further studies are great needed to disentangle relative importance of these two factors."

3) The introduction of previous studies describing marine in-situ produced brGDGTs is too concise. Furthermore, in the discussion I miss how the conclusions from the authors fit with previously published manuscripts? Can we say anything about the water depths at which brGDGTs are produced?

Response: we discussed in more details about in-situ of brGDGTs in the revised manuscript. Please see our response below (line 97-90). To the best of our knowledge, there is no study addressing production of brGDGTs at different water depth. However, a recent study from Kim et al. (2015) has demonstrated an influence of deep water derived iGDGTs on TEX86. So if brGDGTs are also produced in deep water, it would alter brGDGTs' proxies. We discussed this point in section 3.2 "Why do marine sediments have higher IIIa/IIa values than soils?". From line 535 to 543, we said "Alternatively, if brGDGTs in marine sediments are partially derived from deep-water dwelling or benthic organisms, cold deep water (generally 1‒2 oC) would cause higher IIIa/IIa values in marine sediments, as we observed in this study. Although to the best of our knowledge, there is no study reporting in situ production of brGDGTs throughout water column in ocean. Recent studies (Kim et al., 2015; Taylor et al., 2013) have suggested that Thaumarchaeota thriving in the deeper, bathypelagic water-column (>1000 m water depth) biosynthesized iGDGTs with different compositions as surface dwelling Thaumarchaeota, and thereby influences signals of TEX86."

Minor comments: 4ïijĽL14. Rephrase this so the "presumed source" of brGDGTs (soil, peat) is introduced first.

Response: we made change in the revised manuscript as: "Presumed source specificity of branched glycerol dialkyl glycerol tetraethers (brGDGTs) from bacteria thriving in soil/peat and isoprenoid GDGTs (iGDGTs) from aquatic organisms led to the development of several biomarker proxies for biogeochemical cycle and paleoenvironment."

5ïijĽL33: Use 'have been' instead of 'have become'.

Response: we made correction according to reviewer's comment.

6ïijĽL43: Rephrase, this is a confusing sentence. The stereoconfiguration of the glycerol moiety indicates that they are produced by bacteria, not the fact that they are abundant in soils.

Response: We rewrote as "In contrast, the 1,2-di-O-alkyl-sn-glycerol configuration of brGDGTs is interpreted as an evidence for a bacterial rather than archaeal origin for brGDGTs (Sinninghe Damsté et al., 2000; Weijers et al., 2006)" in the revised manuscript.

7) L49: Please include that 15 brGDGT compounds are generally encountered in soils. Besides the variation in the number of methyl groups and cyclopentane moieties, the location of the outer branches has been shown to shift as well.

Response: we accept this suggestion, and rewrote sentences as "So far, only two species of Acidobacteria were identified to contain one brGDGT with two 13,16-dimethyl octacosanyl moieties (Sinninghe Damsté et al., 2011), which is contrast to high diversity and ubiquitous occurrence of 15 brGDGT isomers in environments (De Jonge et al., 2014; Weijers et al., 2007b)."

8) L54: Use Thaumarchaeota instead of crenarchaea.

Response: we made change in the revised manuscript.

9) L58: Here, you can also refer to 'Weijers et al. (2014), Constraints on the sources of branched tetraether membrane lipids in distal marine sediments, OG 72'.

Response: We added the reference of "Weijers et al., 2014" in the revised manuscript.

10) L87-90. As this manuscript discusses brGDGTs produced in marine systems, I would expand a bit more on all studies that have provided evidence for the in-situ production of brGDGTs in the marine environment (instead of just listing them up). Now, only the recent Sinninghe Damste (2016) paper is introduced.

Response: We added more discussion about potential marine-derived brGDGTs. We wrote as "Peterse et al. (2009) compared the brGDGT distribution in Svalbard soils and nearby fjord sediments, and found that concentrations of brGDGTs (0.01–0.20 $\mu$g/g dw) in fjord sediments increased towards the open ocean and the distribution was strikingly different from that in soil. Zhu et al. (2011) examined distributions of GDGTs in surface sediments across a Yangtze River-dominated continental margin, and found evidence for production of brGDGTs in the oxic East China Sea shelf water column and the anoxic sediments/waters of the Lower Yangtze River. At the global scale, Fietz et al. (2012) reported a significant correlation between concentrations of brGDGTs and crenarchaeol (p < 0.01; R2 = 0.57–0.99), suggesting that a common or mixed source for brGDGTs and iGDGTs are actually commonplace in lacustrine and marine settings. More recently, Sinninghe Damsté (2016) reported tetraethers in surface sediments from 43 stations in the Berau River delta (Kalimantan, Indonesia), and this result, combined with data from other shelf systems, supported a widespread biosynthesis of brGDGTs in shelf sediments especially at water depth of 50–300 m. "

11) L 91-99 could be restructured, they are not easy to understand.

Response: We rewrote this paragraph as "However, so far no robust molecular indicator is available for estimating source of brGDGTs in marine environments. Considering this, we conduct a detailed study about GDGTs in three cores from the Bohai Sea which are subject to the Yellow River influence to different degree. Our purpose is to evaluate the source discerning capability of different brGDGT parameters, from which the most sensitive parameter is selected and applied for globally distributed marine sediments and soils to test whether it is valid at the global scale. Our study supplies an important step for improving accuracy of brGDGT-derived proxies and better understanding the

marine carbon cycle and paleoenvironments."

12) L 106: 'the marine carbon cycle'

Response: we made change in revised manuscript.

13) L 112. 'the mean depth is' and 'the Bohai Strait, at the east'.

Response: we made the change according to reviewer's suggestion.

14) L 114: 'the second largest river in the world, concerning sediment load (+reference)'

Response: We rewrote this sentence and added the reference as "Several rivers, including Yellow River, the second largest river in the world in terms of sediment load (Milliman and Meade, 1983), drain into the Bohai Sea with a total annual runoff of $890 \times 108$ m3."

15) L 115: 'One gravity core of 64 cm was: : :'

Response: we made correction in the revised manuscript.

16) L 118: respectively can be removed here.

Response: we already removed "respectively".

17) L 125: If this extraction and separation protocol has been described elsewhere, you can simply refer to this original publication. The same goes for the analysis of the GDGTs on the LC system.

Response: This is a good suggestion. In the revised manuscript, we delete the details about extraction and analysis methods. We started this paragraph as "The detailed procedures for lipid extraction and GDGT analyses were described in previous studies (Ding et al., 2015; Xiao et al., 2015)."

18) L 183: This can be rewritten as: De Jonge et al., 2015, 2016.

Response: We made change.

19) L 206: Can the authors not give an indication at which BIT values (both on the local and global scale) the proportion of marine brGDGT becomes problematic? This would be useful from the viewpoint of palaeoclimate reconstructions.

Response: As we stated in the manuscript, "However, the BIT index itself has no ability to distinguish terrestrial vs. aquatic brGDGTs because brGDGTs and crenarchaea used in this index are thought to be specific for soil organic carbon and marine organic carbon, respectively (Hopmans et al., 2004)". Only the combination of BIT and IIIa/IIa can reveal that when BIT is lower than 0.16, a contribution of marine brGDGTs becomes problematic, which was discussed in section 3.3 and figure 7.

20) L 235-237: Is a repetition of the L 237-242.

Response: we deleted this sentences in the revised manuscript.

21) L282.    It surprises me that the authors indicate here that Crenarchaeota/Thaumarchaeota are the probable producers of marine brGDGTs.  Is there any indication that this would be the case?  Alternatively, I would remove this statement.

Response: We agree with reviewer that more solid evidence is needed to draw such conclusion. Trommer et al. (2009) indeed postulated the existence of distinct crenarchaeota community in the Red Sea due to unusally environmental condition. Considering these facts, we did not specify crenarchaeota in the revised manucript. In stead we rewrote the sentecne as "Under such extreme environment, distinct microbial populations may be developed and produced GDGTs different from that in other marine settings (Trommer et al., 2009)".

22) L303. The argument that continental and marine temperatures are significantly different is put aside much too quickly. Indeed, they are both globally distributed, but the temperature of your water bodies will be much more stable throughout the year (which

has an implication of the production temp as soil-derived brGDGTs are thought to be produced mainly in spring-autumn, especially at sites that are partially frozen throughout the year. Furthermore, if marine brGDGTs are produced at the sediment/water interface, this will of course be much colder than the sea surface temperature. Taking this into account, I doubt that the authors will be able to make a strong case on their proposed absence of a temperature difference between soil and marine brGDGTs.

Response: This is a good comment and already mentioned in major comment. We added detailed discussion in our revised manuscript. Please see our response for Comment #2, particularly about production of GDGTs in surface and deep water with large different temperature.

23) L304. If the authors want to discuss this trend between soil pH and III/II, they have to provide a plot. Does this trend also apply for more extreme pH values? Can it be strengthened by determining which compound causes this trend (IIa, IIa', IIIa, IIIa')?

Response: Good suggestion. We added a figure to show a trend between soil pH and IIIa/IIa (Fig. 6). We agree the separation of 5-methyl and 6-methyl brGDGTs by 2D HPLC-MS may strengthen our hypothesis. However, most available data on brGDGTs did not distinguish these two types of isomers. So we still combine 5-methyl and 6-methyl brGDGTs in current study. But we, along with several groups, are currently using advanced HPLC-MS method to quantify 5-methyl and 6-methyl brGDGTs for more samples. We plan to review 5-methyl and 6-methyl brGDGTs in future when sufficient data are available, but at current stage, this is beyond the scope of this manuscript.

24) L308: The pH of marine water is indeed fairly stable, but it can be very different in pore waters in the sediments. This should be mentioned.

Response: The production of brGDGTs in pore water of sediments cannot be excluded, although they are likely not as important as water column. In the revised manuscript, we added the discussion as "It should be pointed out that unlike fairly stable pH of overlying sea water, the pH of pore waters in marine sediments can vary significantly,

which may influence compositions of brGDGTs. Nevertheless, at current stage, the occurrence of higher IIIa/IIa values in marine sediments is most likely attributed to relatively higher pH and lower deep water temperature. Further studies are needed to disentangle relative importance of these two factors."

25) L367: 'and a compilation of'

Response: we added "a" before compilation.

26) L364-367: I do not agree that the authors have enough evidence and data on this to make this conclusion.

Response: we agree more studies are needed to unambiguously determine source of brGDGTs in marine environments. So in the end of the revised manuscript, we added sentences as "We also note a relatively large scatter of the IIIa/IIa ratio within both terrestrial and marine realms, and different environmental responses of 5-methyl and 6-methyl brGDGTs (e.g., De Jonge et al., 2014, 2016; Xiao et al., 2015). As a result, the separation of these two types of isomers is needed in future studies to develop more accurate brGDGTs' proxies."

27) References: please check the manuscripts guidelines. Journal names are to be abbreviated.

Response: we update the references with abbreviation journal name.

28) General: In the manuscript text, the authors should pay attention to the order of references. Older references should come first.

Response: we reorganized our references according to the requirement of Biogeosciences.
* * *
[Figure]

[Figure]

R$^2$=0.51

pH

IIIa/IIa

**Fig. 1.** Fig. 6 a plot showing a positive correlation between IIIa/IIa and pH

---

## Short Comment (SC1) · 1 Sep 2016

The manuscript is well written. Even most of the data were organized from previous studies, the authors called an attention using IIIa/IIa ratio as a alternative or even better proxy to differentiate soil and marine production of brGDGTs. This manuscript also calls the attention to further look at the the 5-methy and 6-methyl brGDGTs based ratio (IIIa, IIIa', IIa or IIa'). Overall the finding of this manuscript is novel and of great relevance to the scientific communities.Most of the interpretation of the data in this study appears sound to me. I see the merit of this work as a general organic geochemist,

even I am not a GDGT expert. I checked the authors' response to the Anonymous Referee #1 and did see they address them well. There are only a few points of weaknesses identifiable and listed below which needs to be addressed. After that, I wholly support the publication of the manuscript.

Note: I didn't see the updated (corrected) version of manuscript after the authors' response to Anonymous Referee #1 so I am still referring to the original manuscript that I downloaded from http://www.biogeosciences-discuss.net/bg-2016-235/bg-2016-235.pdf

Minor comments

Line 170, when you say significantly is it P < 0.05 or 0.01. Please make this clear. Line 230, 220 kg/m3 (use superscript for 3) Line 238-241, is there any additional evidence to support the different terrestrial organic matter input from your previous work? Such as n-alkane based proxy or triterpenoids etc. It would be great if you can find additional evidence to further support your claim. Line 241-242, it will also be great if you can find additional evidence of land erosion. I know there is a few suit of biomarkers related to land erosion such as some hopanoid series. If you don't have this data available, this could be part of the future work. Line 271, when I wrote "P<0.05", I prefer to use italic script for "P" in order to differentiate the abbreviation of Phosphor Line 279, should be "little" instead of "litter" Line 280, "water salinity up to 41", 41 PSU or ppt, I suggest make this clear. Line 285, you are using 2.6∼5.1, but sometimes you also use 0.4-0.9 (line 269). Please be consistent Line 322, should be written as "IIa/IIa" instead of "IIa/II"

Please check your references because I did see a few typos and non-consistent format

Lines 387, 391, 460, 529, et not Et Line 421, delete "-93" Line 441, "Peng, P.A." not "Peng, P.a." Line 457, delete "-90" Line 467, "and" not "&" Line 513, add " 17" after "Communications" Line 568, delete "-90" Line 681, add a space after "Sparkes et al."

---

## Author Comment (AC2) · 5 Sep 2016

First, we thank Dr. Ding He for commenting our manuscript. His comment is generally positive and acknowledges our work's merit. Meanwhile he provides several suggestions that we found they are useful to improve our manuscript. Here, I address them point by point.

Comment 1) I didn't see the updated (corrected) version of manuscript after the authors' response to Anonymous Referee #1 so I am still referring to the original manuscript that I downloaded from http://www.biogeosciences-discuss.net/bg-2016-

235/bg-2016-235.Pdf. Response: Yes, we did not uploaded the revised manuscript according to the anonymous referee 1. Since we have received the comment from the 2nd reviewer, we uploaded a completely revised manuscript according to the reviewer 1and Ding's comments (please see the supplementary pdf file). All changes were highlighted in the resubmission.

Comment 2: Line 170, when you say significantly is it $P < 0.05$ or 0.01. Please make this clear. Response: Good comment. In the revised manuscript, we rewrote the sentence as "Squared Pearson correlation coefficients ($R^2$) were reported and a significance level is $p < 0.05$."

Comment 3: Line 230, 220 kg/m3 (use superscript for 3) Line 238-241, is there any additional evidence to support the different terrestrial organic matter input from your previous work? Such as n-alkane based proxy or triterpenoids etc. It would be great if you can find additional evidence to further support your claim. Line 241-242, it will also be great if you can find additional evidence of land erosion. I know there is a few suit of biomarkers related to land erosion such as some hopanoid series. If you don't have this data available, this could be part of the future work. Response: we made change according to reviewer's suggestion. We actually analyzed various terrestrial biomarkers such as long chain n-alkanes and C29 sterols, both of which showed more terrestrial OM input in nearshore core than that from central Bohai Sea. However, due to page limitation, we do not present those data. In stead, we added a sentence as "These GDGTs' results, consistent with other terrestrial biomarkers such as C29 and C31 n-alkanes and C29 sterol (data not showed here), strongly suggest that the IIIa/IIa ratio is a sensitive indicator for assessing source of brGDGTs in the Bohai Sea."

Comment 4: Line 271, when I wrote "P<0.05", I prefer to use italic script for "P" in order to differentiate the abbreviation of Phosphor Response: we accepted this suggestion and made change in the revised manuscript.

Comment 5: Line 279, should be "little" instead of "litter" Line 280, "water salinity up to 41", 41 PSU or ppt, I suggest make this clear. Response: we made correction for these spelling errors in the revised manuscript. We also added "PSU" after "41".

Comment 6: Line 285, you are using 2.65∼1, but sometimes you also use 0.4-0.9 (line 269). Please be consistent Line 322, should be written as "IIa/IIa" instead of "IIa/II" Response: we changed "2.65∼1" into "2.65-1" to make consistent throughout the manuscript. We added "a" after "II" in line 322.

Comment 7: Please check your references because I did see a few typos and non-consistent format Lines 387, 391, 460, 529, et not Et Line 421, delete "-93" Line 441, "Peng, P.A." not "Peng, P.a." Line 457, delete "-90" Line 467, "and" not "&" Line 513, add " 17" after "Communications" Line 568, delete "-90" Line 681, add a space after "Sparkes et al." Response: we updated the reference formats in order to make them consistent throughout the manuscript. We uploaded the revised manuscript as supplement.pdf. All changes were highlighted.

Please also note the supplement to this comment:
http://www.biogeosciences-discuss.net/bg-2016-235/bg-2016-235-AC2-
supplement.pdf

**Supplement:**

[revised manuscript text omitted]

---

## Referee Comment (RC2) · Anonymous Referee #2 · 9 Oct 2016

The paper by Xiao et al is a very interesting and valuable contribution to the study of the distribution and origin of branched GDGTs (brGDGT) in mesophilic marine environments and their use as climate proxies. The authors show, initially for the Bohai Sea and subsequently in an extensive data set, that the ratio of two branched GDGTs (termed IIIa/IIa) are dependent on their geographical location. So that values of the ratio in continental sites, coastal environments and marine settings span, generally (as the authors themselves state in multiple occasions), different range of values. This finding, in my opinion, should grant the paper publication in a number of (bio)geochemical journals, as the data and method seems very sound, and the sample set, together with

the use of bibliographic data, are very comprehensive. One should congratulate the authors for such a compilation of data.

The paper is in general well written and presented although there are a number of spelling and grammatical mistakes that still need to be addressed to achieve that the language is fluent and precise.

Any of my concerns with the paper derive from the interpretations and implications that the authors derive from their observations on the distribution of the IIIa/IIa. Some of their interpretations are just hypothesis, but are treated as facts as the paper progresses so that the conclusions contain a number of statements that are not backed up by the data.

Unlike what is claimed in the paper, the authors do not provide proof that in situ production of brGDGTs is actually taking place in marine settings. This is just their hypothesis, albeit plausible, to explain the observed distribution of IIIa/IIa values in the Bohai sediments and other marine settings. Consequently, the paper should be revised to differentiate between actual findings (i.e. IIIa/IIa values), and their proposed interpretation(s) to explain the geographical distribution of the data values. Moreover, it is always useful, when interpreting data, to consider alternative interpretations, if only to discard them and test the strength of the apparently most plausible proposition. For instance, have the authors consider the role of hydrodinamically sorting in explaining IIIa/IIa values in marine settings?. Particles of different sizes may contain different compositions of lipids, and thus sediments change in composition with distance from shore in parallel with changes in particle size.

In this regard, in the title and Conclusion, the authors cannot claim that "in situ production of brGDGTs in marine environments is a ubiquitous phenomenon", because this is an interpretation that they reach to explain why brGDGTs IIIa/IIa ratios have different values in continental, coastal and marine sediments. No other alternative explanations are explored, and no evidence is produced that actual in situ production of

brGDGTs is occurring. At the moment it is a plausible hypothesis, but not a fact. This confusion between hypothetical interpretations and facts is particularly acute in the following statement in the conclusions: "in situ production... is particularly important for those marine sediments with low BIT index (<0.16) where brGDGTs are exclusively of a marine origin." No evidence is produced to demonstrate this statement. Similarly, the authors do not prove that "the IIIa/IIa values... in marine sediments reflects an influence of pH rather than temperature on the biosynthesis of brGDGTs by source organisms.". No data, including measures of pH in the samples, are made available to prove such a claim.

The authors also make extensive use of the BIT index throughout the paper. Given that such an index is used to evaluate the information and possible applications of the IIIa/IIa ratios, the authors should thoroughly discuss the pros and cons, or rather limitations of the BIT index. However, the discussion of this index fails to acknowledge, except as an afterthought in the last line of the paper (li 363), that BIT values are not just dependent on the inputs of soil brGDGTs, but also on the productivity of marine Archaea, which is linked to a large extent to marine productivity. For instance, sites with equal inputs of terrestrial brGDGTS but different local productivity would display different BIT values. In my opinion, the extensive discussion of figure 6, which leads to the generation of figure 7, is meaningless unless proper appraisal is made of what BIT values variability actually means, and proper consideration is made of how changes in productivity influence BIT. It should also be noted that iGDGTs, and chrenarchaeol, are found in soils.

Below some more detailed comments.

Lines 34 and 35: to estimate environmental variables in the past

Line 45: the attribution of the brGDGTs is still hypothetical. But in any case it is unclear why the authors claim that their preferential occurrence in soils/peats means that they are derived from bacteria

Li 48-50: the text should be rewritten, it is unclear what the authors are trying to say

Li 58-59: the BIT index is used for what?

Li 58-60: in here the authors should also comment the often overlooked drawback of the BIT index, namely that is dependent on the input of chrenarcheol, which is linked to marine productivity. Consequently, BIT values are not just dependent on the inputs of soil brGDGTs, but also on the productivity of marine Archaea. For instance, sites with equal inputs of terrestrial brGDGTS but different local productivity would display different BIT values. There are a number of references out there discussing this issue, for instance:

*Herfort, L., S. Schouten, J. P. Boon, M. Woltering, M. Baas, J. W. H. Weijers, and J. S. Sinninghe Damsté (2006), Characterization of transport and deposition of terrestrial organic matter in the southern North Sea using the BIT index, Limnol. Oceanogr., 51, 2196–2205, doi:10.4319/lo.2006. 51.5.2196.

*Fietz, S., Martínez-Garcia, A., Huguet, C., Rueda, G., & Rosell-Melé, A. (2011). Constraints in the application of the Branched and Isoprenoid Tetraether index as a terrestrial input proxy. Journal of Geophysical Research, 116(C10), 1–9.

*Smith, R. W., Bianchi, T. S., & Savage, C. (2010). Comparison of lignin phenols and branched/isoprenoid tetraethers (BIT index) as indices of terrestrial organic matter in Doubtful Sound, Fiordland, New Zealand. Organic Geochemistry, 41(3), 281–290. http://doi.org/10.1016/j.orggeochem.2009.10.009

Li 76: "The premise of all brGDGT", do you the authors mean: the underlying assumption?

Li 79: "supporting in situ production of brGDGTs": the authors cited hypothesized the occurrence of in situ production, so their studies supported the hypothesis of the occurrence of...

Li 89: instead of "supported" use "findings were coherent with the hypothesis that

brGDGTs are in situ produced in marine environaments".

Li 91: instead of "river" use "fluvial inputs or run off"

Li 93-94: brGDGTs have not been analyzed in that many dust samples to date, but it may be obvious to assume in the meantime that their concentration in dust will be as high, proportionally, to the contents of soil particles in dust. In this section it is relevant to cite as well the just published paper by Yamamoto et al., 2016, GCA, 191, 15 October 2016, Pages 239–254.

Li 95: "became": why just in the past?

Li 112: mean depth

Li 115-116: One 64 cm long gravity core

Li 117: namely?

Li 121-122: I would rephrase "cores cover the sedimentation period of less than 100 years"

Li 125: samples were ground with a mortar and pestle

Li 137: Define "EtOAc"

Li 138; I would rephrase "Samples were injected...", where?

Li 139: As this is relatively novel, I would indicate from which reference(s) the HPLC method is derived.

Li 175: "The dataset in this study are composed of GDGTs from.." absolute/relative concentrations?, fluxes?

Li 177: I would rephrase "and have water depth"

Li 197: I would rewrite "Both iGDGTs including crenarchaea and brGDGTs". Chrenarchaea or chrenarcheaol?

Li 206: "expectable"?

Li 207-210: iGDGTs are found in soils too.

Li 208: the statement does not make much sense as the BIT was not "designed" for this purpose as it has already been discussed

Li 212-214: I would rephrase this section "all parameters except MI can distinguish Chinese soils from Bohai Sea sediments"

Li 234-235: "enhanced IIIa/IIa values in the Bohai Sea sediments is caused by in 234 situ production of brGDGTs." The statement should be rephrased to differentiate between actual findings (i.e. IIIa/IIa values), and their proposed interpretation(s) (i.e. in situ production).

Li 237-239: "The site M1 is adjacent to the Yellow River mouth and receives the largest amount of terrestrial organic matter, causing lower IIIa/IIa values". Again, the authors should rephrase the statement to indicate which is their interpretation of the IIA/IIa values, as they do not prove what causes the lower IIIa/IIa values. The same applies to text in lines 262, 272-273, 315, 371-374.

Li 240: "comprises of the least amount of terrestrial organic matter", please justify this statement

Li 242: "strongly", why?, is this a subjective claim or is backed up by some stats.?

Li 246-247: The authors should indicate that they try to validate the ratio as a proxy for something, not to validate the ratio itself, or are they also trying to assess if the IIa and IIIa are ubiquituous?

Li 258: compiling or compilation?

Li 259: brGDGTs concentrations?, fluxes? Data?

Li 274-275: I would rephrase this section

[Figure]

Li 275: Where is the statistical analysis?

Li 278: "unusually low" in which context are they low?

Li 279: "Bab el Mandeb" strait

Li 279: "litter" or low?

Li 280: salinity, no units?

Li 281-283: The Red Sea is an extreme environment?, the authors do not explain why the ratios from environments as different as those in Fig. 5 (e.g. Arctic, Mediterranean, Chilean margin, South China Sea, river waters and soils) fit within the scheme proposed to interpret the IIIa/IIa ratios, whereas the Red Sea does not. The interpretation proposed is not very convincing, particularly as they seem to argue through the text that the producers of brGDGTs in soils and marine settings are not the same type of organisms.

Li 284-286: level or values?

Li 290: "Why do soils have lower IIIa/IIa" and the Red Sea?

Li 302-303: Please explain further what is meant by and why is not related to the IIIa/IIa ratio: "because both soils and marine sediments are globally distributed and their temperatures (MAT vs. sea surface temperature) have no systematic difference".

Li 305: "positive correlation with soil pH (R2=0.43)", really?, with such a R2 value?

Li 305-312: I would use more caution in this section as most of the evidence used to back the authors' interpretation is hypothetical

Li 322-324: the regression in Fig. 6 is the product of wishful thinking. One can fit any curve to a group of unrelated data point and get "satisfactory" R2 value. I think that it is evident from Figure 6 that BIT and the III/II ratios are unrelated. There are two cluster of data. Why samples with BIT values below 0.3 (which are supposed to be only typical

of sites with low terrigenous inputs) have such an spread of III/II values?, Similarly, how come that values of III/II below 0.8, which are proposed to be only found in soils (li 285) has such an spread of BIT values from 0 to 1. It does not make sense to me if both indicators are indicators of marine vs. terrigenous organic carbon. Should not they fit into a simple straightforward linear regression if IIIa/IIa and BIT are both indexes for assessing soil organic carbon (inputs) in marine settings, as claimed by the authors? .

Li 366-368: it is not necessary to say in the conclusions section that the authors have reached some conclusions. It is redundant.

Li 369: Please define what is meant by "generally lower", as it stands it is a subjective statement which is followed by values that are purported to reflect objective thresholds (which are not in fact).

Li 369-370: The authors have not demonstrated the occurrence of terrestrial inputs in all samples studied (e.g. Fig. 6). They cannot claim that high values of III/II occur in sediments "devoid of significant terrestrial inputs". What is meant by significant anyhow?.

Fig. 1: m/z of chrenarchaeol?

Fig. 4 combines 4 graphs extracted from published papers that are unrelated to each other, and I think that they should go in different figures for coherence sake in the supplementary information section. Explain the abbreviations in the x-axis in fig. 4b in the legend.

Fig.5. The use of symbols of different size prevents the visualization of all the data in the map, as the big dots cover smaller dots, and also are easier to visualize that smaller dots, giving the impression that "there are more of them". Please use another way of visualizing all the data that provides equal weight to data with different range of values.

Table 1: where are the samples from?. Please explain further what is mean by: "Different letters (a, b, c, d) represent significant difference at the level of p<0.05."

[Figure]

---

## Author Comment (AC3) · 13 Oct 2016

On behalf of my coauthor, I really appreciate the reviewer 2 supplying very detailed comments, which are helpful to improve our manuscript. Overall, the reviewer 2 acknowledge the merit of our paper. As he or she said that "the work by Xiao et al. is a very interesting and valuable contribution to the study of the distribution and origin of branched GDGTs (brGDGT) in mesophilic marine environments and their use as climate proxies" and "This finding, in my opinion, should grant the paper publication in a number of (bio)geochemical journals". The main concern of the reviewer 2

is not 100% sure of in situ brGDGTs in aquatic environments. In our opinion, more and more studies support that brGDGTs are also biosynthesized by aquatic organisms. Presently, the in-situ production of brGDGTs in aquatic environments has been accepted by most organic geochemists. For examples, Peterse et al. (2009a), Zhu et al. (2011), Liu et al. (2014) Weijers et al. (2014) and Zell et al. (2014) all observed different brGDGT compositions between marine sediments from different seas and soils on adjacent lands, so they proposed in situ production of brGDGTs in marine environments. Similar conclusions have been reached for lacustrine settings by Sinninghe Damsté et al. (2009), Tierney & Russell (2009) and Tierney et al. (2012) as well as for rivers by Zhu et al. (2011), De Jonge et al. (2015), French et al. (2015) and Zell et al., (2015). Most those studies were published after 2010, so more and more organic geochemists accepted the view that brGDGTs were produced in aquatic environments. Peterse et al. (2009) compared the brGDGT distribution in Svalbard soils and nearby fjord sediments, and found that concentrations of brGDGTs (0.01–0.20 $\mu$g/g dw) in fjord sediments increased towards the open ocean and the distribution was strikingly different from that in soil. Zhu et al. (2011) examined distributions of GDGTs in surface sediments across a Yangtze River-dominated continental margin, and found evidence for production of brGDGTs in the oxic East China Sea shelf water column and the anoxic sediments/waters of the Lower Yangtze River. At the global scale, Fietz et al. (2012) reported a significant correlation between concentrations of brGDGTs and crenarchaeol (p < 0.01; R2 = 0.57–0.99), suggesting that a common or mixed source for brGDGTs and iGDGTs are actually commonplace in lacustrine and marine settings. Weijers et al. (2014) found that distributions of African dust-derived brGDGTs were similar to those of soils but different from those of distal marine sediments, providing a possibility to distinguish terrestrial vs. marine brGDGTs based on molecular compositions. More recently, Sinninghe Damsté (2016) reported tetraethers in surface sediments from 43 stations in the Berau River delta (Kalimantan, Indonesia), and this result, combined with data from other shelf systems, are coherent with the hypothesis that brGDGTs are in situ produced in shelf sediments especially at water depth of 50–

m. In the remote ocean where no direct impact from land erosion via rivers takes place, eolian transport and in situ production are major contributors for brGDGTs. So in our opinion, based on recent studies and our study, the in-situ production of brGDGTs is a ubiquitous phenomenon in marine environments, as our title states. We try our best to address all comments point by point as follows. In addition, all changes are highlighted in the revised manuscript.

Comment: Lines 34 and 35: to estimate environmental variables in the past Response: we accept this suggestion and rewrote the sentence as "These lipids have been a focus of attention of organic geochemists for more than ten years because they can be used to estimate environmental variables in the past such as temperature, soil pH, organic carbon source and microbial community structure"

Comment: Line 45: the attribution of the brGDGTs is still hypothetical. But in any case it is unclear. why the authors claim that their preferential occurrence in soils/peats means that they are derived from bacteria Response: the source assignment of branched GDGTs to bacteria is based on structural configuration (1,2-di-O-alkyl-sn-glycerol configuration). It is generally accepted by Organic Geochemists.

Comment: Li 48-50: the text should be rewritten, it is unclear what the authors are trying to say Response: in revised manuscript, we rewrote the sentences as: "So far, only one brGDGT with two 13,16-dimethyl octacosanyl moieties was unambiguously detected in two species of Acidobacteria (Sinninghe Damsté et al., 2011), which hardly explains high diversity and ubiquitous occurrence of up to 15 brGDGT isomers in environments (Weijers et al., 2007b; De Jonge et al., 2014). Therefore, other biological sources of brGDGTs, although not yet identified, are likely."

Comment: Li 58-59: the BIT index is used for what? Response: we rewrote the sentence as "Since its advent, the BIT index has been increasingly used to trace soil organic matter in different environments" in revised manuscript.

Comment: Li 58-60: in here the authors should also comment the often overlooked

drawback of the BIT index, namely that is dependent on the input of chrenarcheol, which is linked to marine productivity. Consequently, BIT values are not just dependent on the inputs of soil brGDGTs, but also on the productivity of marine Archaea. For instance, sites with equal inputs of terrestrial brGDGTS but different local productivity would display different BIT values. There are a number of references out there discussing this issue, for instance: *Herfort, L., S. Schouten, J. P. Boon,M. Woltering, M. Baas, J. W. H. Weijers, and J. S. Sinninghe Damsté (2006), Characterization of transport and deposition of terrestrial organic matter in the southern North Sea using the BIT index, Limnol. Oceanogr., 51, 2196–2205, doi:10.4319/lo.2006. 51.5.2196. *Fietz, S., Martínez-Garcia, A., Huguet, C., Rueda, G., & Rosell-Melé, A. (2011). Constraints in the application of the Branched and Isoprenoid Tetraether index as a terrestrial input proxy. Journal of Geophysical Research, 116(C10), 1–9. *Smith, R. W., Bianchi, T. S., & Savage, C. (2010). Comparison of lignin phenols and branched/isoprenoid tetraethers (BIT index) as indices of terrestrial organic matter in Doubtful Sound, Fiordland, New Zealand. Organic Geochemistry, 41(3), 281–290. http://doi.org/10.1016/j.orggeochem.2009.10.009 Response: This is a good comment. In the revised manuscript, we added this comment as well as related references. We wrote as "However, the BIT index is not just dependent on the abundance of brGDGTs, which reflects the input of soil organic matter, but also on the abundance of chrenarcheol, which is linked to marine productivity (e.g., Herfort et al., 2006ïijŽSmith et al, 2010; Fietz et al., 2011)."

Comment: Li 76: "The premise of all brGDGT", do you the authors mean: the underlying assumption? Response: Yes, that is what we meant. In order to avoid confusion, we changed "premise" into "underlying assumption" in the revised manuscript.

Comment: Li 79: "supporting in situ production of brGDGTs": the authors cited hypothesized the occurrence of in situ production, so their studies supported the hypothesis of the occurrence of... Response: Actually, there are a number of studies supporting in-situ production of brGDGTs aquatic environments such as lake, river and marginal seas, such as Svalbard, Norway(Peterse et al., 2009a, OG); Yangtze River and estuary (Zhu et al., 2011, OG); Black Sea and Cariaco Basin (Liu et al., 2014, Mar Chem); equatorial West African coast (Weijers et al., 2014, OG); Portuguese margin (Zell et al., 2014; Biogeosciences). So the in-situ production of brGDGTs is generally accepted by organic geochemists.

Comment: Li 89: instead of "supported" use "findings were coherent with the hypothesis that brGDGTs are in situ produced in marine environaments". Response: Although we are pretty sure for the existence of in-situ production of brGDGTs in aquatic environments (see our response above), we rewrote as "this result, combined with data from other shelf systems, are coherent with the hypothesis that brGDGTs are in situ produced in shelf sediments especially at water depth of 50–300 m."

Comment: Li 91: instead of "river" use "fluvial inputs or run off" Response: we accepted this comment and changed "river" into "fluvial inputs"

Comment: Li 93-94: brGDGTs have not been analyzed in that many dust samples to date, but it may be obvious to assume in the meantime that their concentration in dust will be as high, proportionally, to the contents of soil particles in dust. In this section it is relevant to cite as well the just published paper by Yamamoto et al., 2016, GCA, 191,15 October 2016, Pages 239–254.

Comment: Li 95: "became": why just in the past? Response: we change "became" into "are major contributors for brGDGTs".

Comment: Li 112: mean depth Response: we already corrected this spelling mistake.

Comment: Li 115-116: One 64 cm long gravity core Response: we accepted this suggestion and made change in revised manuscript.

Comment: Li 117: namely? Response: we deleted "namely" and rewrote the sentence as "while other two cores, M3 (38.66°N, 119.54°E; 53 cm long) and M7 (39.53°N, 120.46°E; 60 cm long), were collected in July 2013".

Comment: Li 121-122: I would rephrase "cores cover the sedimentation period of less than 100 years" Response: We rewrote the sentence as "showing that the age of the bottom sediments was less than 100 years old".

Comment: Li 125: samples were ground with a mortar and pestle Response: Based on this suggestion and comment from reviewer 1, we rewrote as "The detailed procedures for lipid extraction and GDGT analyses were described in previous studies (Ding et al., 2015; Xiao et al., 2015). Briefly, the homogenous freeze-dried samples were ultrasonically extracted with dichloromethane (DCM)/methanol (3:1 v:v)."

Comment: Li 137: Define "EtOAc" Response: We deleted EtOAc in the revised manuscript, and rewrote the part of analytical procedure in revised manuscript.

Comment: Li 138; I would rephrase "Samples were injected: : :", where? Response: We rewrote the part of Lipid Extraction and Analyses according to the reviewers' comments. Please see the revised manuscript for details (Section 2.2).

Comment: Li 139: As this is relatively novel, I would indicate from which reference(s) the HPLC method is derived. Response: In the beginning of the paragraph, we added the references as "The detailed procedures for lipid extraction and GDGT analyses were described in previous studies (Ding et al., 2015; Xiao et al., 2015)".

Comment: Li 175: "The dataset in this study are composed of GDGTs from.." absolute/relative concentrations?, fluxes? Response: We changed into "The dataset in this study are composed of relative abundance of GDGTs and derived parameters from 1354 globally distributed soils and 589 marine sediments (Fig. 2 and supplementary data)" in the revised manuscript.

Comment: Li 177: I would rephrase "and have water depth" Response: we changed into "and the water depth ranges from 1.0 to 5521 m".

Comment: Li 197: I would rewrite "Both iGDGTs including crenarchaea and brGDGTs". Chrenarchaea or chrenarcheaol? Response: We rewrote as "A series of iGDGTs including crenarchaeol and brGDGTs including 5-methyl and 6-methyl isomers were detected in Bohai Sea sediments." In the revised manuscript.

Comment: Li 206: "expectable"? Response: we changed into "Such difference is not surprising".

Comment: Li 207-210: iGDGTs are found in soils too. Response: It is true iGDGTs are present in soils at low abundance. So we rewrote the sentence as "However, the BIT index itself has no ability to distinguish terrestrial vs. aquatic brGDGTs because brGDGTs and crenarchaeol used in this index are thought to be specific for soil organic carbon and marine organic carbon, respectively (Hopmans et al., 2004), although crenarchaeol is also present in soils at low abundance (Weijers et al., 2006)."

Comment: Li 208: the statement does not make much sense as the BIT was not "designed" for this purpose as it has already been discussed Response: We agree on that BIT is designed for estimation of terrestrial organic matter in aquatic environments in Hopmans et al. (2004). However, more and more recent studies suggest that branched GDGTs are also in situ products in aquatic environments, such as lake, river and seas. Although the BIT index is still useful for two end-members (aquatic and soil organic matter), but it lack the capability to discern the source of branched GDGTs from terrestrial or in situ aquatic origin in seas. In contrast, the character of brGDGTs' composition (excluding crenarchaeol) can be used to estimate the source of brGDGTs, such as IIIa/IIa ratio proposed in our study. Nevertheless, we rewrote the sentence here as "However, the BIT index itself has no ability to determine the source of brGDGTs (terrestrial vs. aquatic) because brGDGTs and crenarchaeol used in this index are thought to be specific for soil organic carbon and marine organic carbon, respectively (Hopmans et al., 2004), although crenarchaeol is also present in soils at low abundance (Weijers et al., 2006)."

Comment: Li 212-214: I would rephrase this section "all parameters except MI can distinguish Chinese soils from Bohai Sea sediments" Response: we accept this suggestion and rewrote this section. In the revised manuscript, we rewrote as "We performed ANOVA for a variety of brGDGTs' parameters. All results except from MI show a significant difference between Chinese soils and Bohai Sea sediments. The IIIa/IIa ratio is the most sensitive parameter which can completely separate the samples into four groups, Chinese soils ($0.39 \pm 0.25$; Mean$\pm$SD; same hereafter), M1 sediments ($0.63 \pm 0.06$), M3 sediments ($1.16 \pm 0.12$) and M7 sediments ($0.93 \pm 0.07$)."

Comment: Li 234-235: "enhanced IIIa/IIa values in the Bohai Sea sediments is caused by in 234 situ production of brGDGTs." The statement should be rephrased to differentiate between actual findings (i.e. IIIa/IIa values), and their proposed interpretation(s) (i.e. in situ production). Response: It

Comment: Li 237-239: "The site M1 is adjacent to the Yellow River mouth and receives the largest amount of terrestrial organic matter, causing lower IIIa/IIa values". Again, the authors should rephrase the statement to indicate which is their interpretation of the IIA/IIa values, as they do not prove what causes the lower IIIa/IIa values. The same applies to text in lines 262, 272-273, 315, 371-374. Li 240: "comprises of the least amount of terrestrial organic matter", please justify this statement. Li 242: "strongly", why?, is this a subjective claim or is backed up by some stats.? Response: Good comment. We rewrote the sentences as "In our study, the site M1 is adjacent to the Yellow River mouth and receives the largest amount of terrestrial organic matter, causing lower IIIa/IIa values ($0.63 \pm 0.06$). In contrast, the site M3 located in central Bohai Sea comprises of the least amount of terrestrial organic matter, resulting in higher IIIa/IIa values ($1.16 \pm 0.12$). The intermediate IIIa/IIa values at the site M7 ($0.93 \pm 0.07$) is attributed to moderate land erosion nearby northern Bohai Sea (Fig. 2). These GDGTs' results, consistent with other terrestrial biomarkers such as C29 and C31 n-alkanes and C29 sterol (data not showed here), suggest that the higher IIIa/IIa values in the Bohai Sea sediments compared to Chinese soils ($0.39 \pm 0.25$) is most likely caused by in situ production of brGDGTs."

Comment: Li 246-247: The authors should indicate that they try to validate the ratio as a proxy for something, not to validate the ratio itself, or are they also trying to assess if the IIa and IIIa are ubiquituous? Response: We added the sentence to state this point. In the beginning of this paragraph, we wrote as "Similar to Bohai Sea in this study, the compounds brGDGT IIa and IIIa are also ubiquitously present in these environments."

Comment: Li 258: compiling or compilation? Response: We changed "compiling" into "Compilation".

Comment: Li 259: brGDGTs concentrations?, fluxes? Data? Response: It is the concentration of brGDGTs. We made change in revised manuscript.

Comment: Li 274-275: I would rephrase this section. Where is the statistical analysis? Response: we wrote as "We further extend the dataset on global scale (Fig. 5), showing that the IIIa/IIa ratio is still significantly higher in marine sediments than soils (p<0.01)." Similar to Bohai Sea, we also perform statistical analysis but detailed results are not shown here.

Comment: Li 278: "unusually low" in which context are they low? Response: Red Sea sediments showed unusually low IIIa/IIa values compared to other marine sediments. So we changed the sentence as "An exception was observed for Red Sea sediments which have unusually low IIIa/IIa values (0.39±0.21) compared to other marine sediments (>0.87)."

Comment: Li 279: "Bab el Mandeb" strait Response: We corrected it.

Comment: Li 279: "litter" or low? Response: we corrected this spelling error.

Comment: Li 280: salinity, no units? Response: We added "PSU" after salinity.

Comment: Li 281-283: The Red Sea is an extreme environment?, the authors do not explain why the ratios from environments as different as those in Fig. 5 (e.g. Arctic, Mediterranean, Chilean margin, South China Sea, river waters and soils) fit within the scheme proposed to interpret the IIIa/IIa ratios, whereas the Red Sea does not. The interpretation proposed is not very convincing, particularly as they seem to argue

through the text that the producers of brGDGTs in soils and marine settings are not the same type of organisms. Response: we explained the reasons why Read Sea is different in brGDGTs' compositions from other marine systems. In the manuscript, we wrote as "the Red Sea has a restricted connection to the Indian Ocean via the Bab el Mandeb Strait. This, combined with high insolation, low precipitation and strong winds result in surface water salinity up to 41 PSU in the south and 36 PSU in the north of the Red Sea (Sofianos et al., 2002). Under such extreme environment, distinct microbial populations may be developed and produced GDGTs different from that in other marine settings (see Trommer et al., 2009 for details).

Comment: Li 284-286: level or values? Response: we changed level into values in the revised manuscript.

Comment: Li 290: "Why do soils have lower IIIa/IIa" and the Red Sea? Response: As we mentioned above, Red Sea is an exception. So we add "generally" in the sentence. Considering the comments from reviewer 1 and 2, we rewrote as "Why do marine sediments generally have higher IIIa/IIa values than soils?"

Comment: Li 302-303: Please explain further what is meant by and why is not related to the IIIa/IIa ratio: "because both soils and marine sediments are globally distributed and their temperatures (MAT vs. sea surface temperature) have no systematic difference". Response: We accepted reviewer 1 and 2's suggestions, and rewrote the whole paragraph. Pleases see line 352 to 389 for details.

Comment: Li 305: "positive correlation with soil pH (R2=0.43)", really?, with such a R2 value? Response: we already double check this point. In the revised manuscript, we added figure 6 that shows even higher R2 value between the IIIa/IIa ratio and soil pH (R2=0.51; Fig. 6) when global soil dataset are included.

Comment: Li 305-312: I would use more caution in this section as most of the evidence used to back the authors' interpretation is hypothetical Response: we added this content in the revised manuscript as: "It should be pointed out that unlike fairly stable pH

of overlying sea water, the pH of pore waters in marine sediments can vary significantly, which may influence compositions of brGDGTs. Nevertheless, at current stage, the occurrence of higher IIIa/IIa values in marine sediments is most likely attributed to relatively higher pH and lower deep water temperature. Further studies are needed to disentangle relative importance of these two factors."

Comment: Li 322-324: the regression in Fig. 6 is the product of wishful thinking. One can fit any curve to a group of unrelated data point and get "satisfactory" R2 value. I think that it is evident from Figure 6 that BIT and the III/II ratios are unrelated. There are two cluster of data. Why samples with BIT values below 0.3 (which are supposed to be only typical of sites with low terrigenous inputs) have such an spread of III/II values?, Similarly, how come that values of III/II below 0.8, which are proposed to be only found in soils (li 285) has such an spread of BIT values from 0 to 1. It does not make sense to me if both indicators are indicators of marine vs. terrigenous organic carbon. Should not they fit into a simple straightforward linear regression if IIIa/IIa and BIT are both indexes for assessing soil organic carbon (inputs) in marine settings, as claimed by the authors? . Response: we tried the different correlation between IIIa/IIa and BIT. The exponential correlation has much higher R2 value than linear correlation. This is the reason we showed exponential curve in figure 6 (figure 7 in revised manuscript). But more important, our key point is not correlation between IIIa/IIa and BIT. We want to show different clusters of marine and terrestrial samples based on these two indicators. We chose <0.3 and >0.67 as threshold values because they can include 90% of samples. We highlight this point in the manuscript.

Comment: Li 366-368: it is not necessary to say in the conclusions section that the authors have reached some conclusions. It is redundant. Response: we delete this redundant sentences in revised manuscript.

Comment: Li 369: Please define what is meant by "generally lower", as it stands it is a subjective statement which is followed by values that are purported to reflect objective thresholds (which are not in fact). Response: we changed

Comment: Li 369-370: The authors have not demonstrated the occurrence of terrestrial inputs in all samples studied (e.g. Fig. 6). They cannot claim that high values of III/II occur in sediments "devoid of significant terrestrial inputs". What is meant by significant anyhow?. Response: Good comment. We rewrote as "Our investigation for brGDGTs in three Bohai Sea cores and globally distributed soils and marine sediments shows that the brGDGTs IIIa/IIa ratio is lower than 0.59 in 90% of soils, but higher than 0.92 in 90% of marine sediments devoid of significant terrestrial inputs"

Comment: Fig. 1: m/z of chrenarchaeol? Response: We confirmed as crenarchaeol

Comment: Fig. 4 combines 4 graphs extracted from published papers that are unrelated to each other, and I think that they should go in different figures for coherence sake in the supplementary information section. Explain the abbreviations in the x-axis in fig. 4b in the legend. Response: we combined the comments from reviewers 1 and 2 as well as Dr. Ding He. In order to express more clearly, we still leave the figure 4 in the manuscript.

Comment: Fig.5. The use of symbols of different size prevents the visualization of all the data in the map, as the big dots cover smaller dots, and also are easier to visualize that smaller dots, giving the impression that "there are more of them". Please use another way of visualizing all the data that provides equal weight to data with different range of values. Response: we asked the suggestions from several senior scientists and biogeochemists. They think it is useful to use different size of symbol to reflect high or low values of IIIa/IIa. So we did not make any change in the revised manuscript.

Comment: Table 1: where are the samples from?. Please explain further what is mean by: "Different letters (a, b, c, d) represent significant difference at the level of $p<0.05$." Response: We added more explanation in the title of Table 1.

Please also note the supplement to this comment:
http://www.biogeosciences-discuss.net/bg-2016-235/bg-2016-235-AC3-
supplement.pdf

[Figure]

**Supplement:**

[revised manuscript text omitted]